



# Effects of upper mantle heterogeneities on lithospheric stress field and dynamic topography

Anthony Osei Tutu[1,2], Bernhard Steinberger[1,3], Stephan V. Sobolev[1,2], Irina Rogozhina[4,1], and Anton A. Popov[5]

[1]Section 2.5 Geodynamic modelling, GFZ German Research Centre for Geosciences, Potsdam, Germany
[2]Institute of Earth and Environmental Science, University of Potsdam, Potsdam, Germany.
[3]Centre for Earth Evolution and Dynamics, University of Oslo, Oslo, Norway.
[4]MARUM Centre for Marine Environmental Sciences, University of Bremen, Bremen, Germany.
[5]Institute of Geosciences, Johann Gutenberg University, Mainz, Germany.

*Correspondence to:* Anthony Osei Tutu (oseitutu@gfz-potsdam.de)

**Abstract.** The orientation and tectonic regime of the observed crustal/lithospheric stress field contribute to our knowledge of different deformation processes occurring within the Earth's crust and lithosphere. In this study, we analyze the influence of the thermal and density structure of the upper mantle on the lithospheric stress field and topography. We use a 3D lithosphere-asthenosphere numerical model with power-law rheology, coupled to a spectral mantle flow code at 300 km depth. Our results

5  are validated against the World Stress Map 2016 and the observation-based residual topography. We derive the upper mantle thermal structure from either a heat flow model combined with a sea floor age model (TM1) or a global S-wave velocity model (TM2). We show that lateral density heterogeneities in the upper 300 km have a limited influence on the modeled horizontal stress field as opposed to the resulting dynamic topography that appears more sensitive to such heterogeneities. There is hardly any difference between the stress orientation patterns predicted with and without consideration of the heterogeneities

10  in the upper mantle density structure across North America, Australia, and North Africa. In contrast, we find that the dynamic topography is to a greater extent controlled by the upper mantle density structure. After correction for the chemical depletion of continents, the TM2 model leads to a much better fit with the observed residual topography giving a correlation of 0.51 in continents, but this correction leads to no significant improvement in the resulting lithosphere stresses. In continental regions with abundant heat flow data such as, for instant, Western Europe, TM1 results in relatively a small angular misfits of 18.30

15  ° between the modeled and observation-based stress field compared 19.90 ° resulting from modeled lithosphere stress with s-wave based model TM2.





# 1 Introduction

The stresses building up in the rigid outermost layer of the Earth are the result of both shallow and deep geological processes. Lithosphere dynamics are defined by a combination of plastic, elastic and viscous flow properties of the lithospheric material (Burov, 2011; Tesauro et al., 2012), while the evolution of the sub-lithospheric mantle is predominantly driven by viscous flow

(Davies, 1977; Forte and Mitrovica, 2001; Steinberger and Calderwood, 2006). This is evident from surface expressions of different deformation processes around the globe, such as for example the ongoing crustal deformation processes that formed the Tibetan Plateau due to the continental collision of the Indian and Eurasian Plates (van Hinsbergen et al., 2011) or the rifting of the African Plate induced by its interaction with the Afar plume head (Ebinger and Sleep, 1998). It has been shown that shallow processes influence both the magnitude and orientation of the lithospheric stresses. Among such processes the most

important are the slab pull, ridge push, trench and continental collision (deformation) (Reynolds et al., 2002) as well as the cratonic root resistance (Naliboff et al., 2012). Also, gravitational effects due to lateral density heterogeneities in the lithosphere and tractions from the mantle flow at the base of the moving plates play an important role. Superposition of different tectonic forces creates dissimilar orientations and regimes of the lithospheric stress field in different regions, as shown by the World Stress Map project (Bird and Li, 1996; Heidbach and Höhne, 2007; Heidbach et al., 2008, 2016).

Furthermore, on a global scale, the intra-plate stress orientation follows a specific pattern at a longer wavelength due to a large force contribution from the convecting mantle (Steinberger et al., 2001; Lithgow-Bertelloni and Guynn, 2004). This first-order stress pattern (long wavelength) is dynamically supported, as the controlling forces correlate well with the forces driving the plate motion in most continental areas such as North and South Americas and Europe (Solomon et al., 1980; Richardson, 1992; Zoback, 1992). Ghosh and Holt (2012) and Steinberger et al. (2001) used different approaches to show that the contribution

of the crust (shallow density structures) to the overall lithospheric stress pattern is rather small compared to that of the mantle buoyancy forces, amounting to ~10 %, except for regions characterized by high altitudes, especially the Tibetan Plateau, where the contribution is larger. In these previous modeling studies, the effect of the crust was determined separately by computing the gravitational potential energy from a crust model, which was subsequently applied as a correction (Steinberger et al., 2001; Ghosh et al., 2013; Ghosh and Holt, 2012). The contribution of the crust with a shallow lithospheric density contrast generates

the second-order pattern (mid-to-short wavelength) in the stress field, mostly coming from topography and the crust isostasy (Zoback, 1992; Zoback and Mooney, 2003; Bird et al., 2006).

At longer wavelengths, the vertical component of the stress field tensor originating from the thermal convection of the mantle rocks (Pekeris, 1935; Steinberger et al., 2001) contributes to the topographic signal. This generates a high dynamic topography in regions of upwelling over the African and Pacific Large Low Shear Velocity Provinces (LLSVP) and low topography above

downwelling in the regions of subduction (Hager and O'Connell, 1981; Hager et al., 1985). On the other hand, at a mid-to-short wavelength, topographic features are influenced by processes such as plume-lithosphere interaction (Lithgow-Bertelloni and Silver, 1998; Thoraval et al., 2006; Dannberg and Sobolev, 2015) and small-scale convection in the upper mantle (Marquart and Schmeling, 1989; King and Ritsema, 2000; Hoggard et al., 2016). However, the largest fraction of topography is caused by isostasy due to variations in crustal thickness and density, as well as density variations in the subcrustal lithosphere.



A number of studies (Čadek and Fleitout, 2003; Forte and Mitrovica, 2001; Garcia-Castellanos and Cloetingh, 2011; Ghosh and Holt, 2012; Steinberger et al., 2001) have presented numerical simulations of different geophysical processes and compared their model results with observations of the lithosphere stress field, dynamic geoid, plate motion velocity and dynamic topography to better understand what processes control these surface observables. For instance, the modeled dynamic geoid

typically gives a good correlation with observations, due to a large contribution of the lower mantle (Čadek and Fleitout, 2003; Hager et al., 1985; Richards and Hager, 1984), but is sensitive to the choice of the mantle viscosity (Thoraval and Richards, 1997). However, the correlation between the modeled dynamic and residual topography is typically found to be weak (Heine et al., 2008; Flament et al., 2012; Steinberger and Calderwood, 2006; Steinberger, 2016; Hoggard et al., 2016). The residual topography is here defined as the observed topography corrected for the variations in the crustal and lithosphere thickness and

density variations and for subsidence of the sea floor with age. One of the reasons for dissimilarities between the modeled and observed topography is our insufficient knowledge of the petrological properties of the upper mantle (Cammarano et al., 2011), for example in relation to the chemical depletion of cratons in continental regions. This is further amplified by the uncertainties in the complex rheological and density structure of the upper mantle due to a wide range of thermal regimes associated with cold subducting plates and cratons, hot plumes and small-scale convection cells (Ebinger and Sleep, 1998; Thoraval et al.,

2006). Another reason is linked to the deficiencies of the state-of-the-art seismic tomography models that often fail to provide the necessary detail about the density/thermal heterogeneities in the upper mantle. Finally, crustal models (e.g. Laske et al., 2013) used to compute the observation-based residual topography are not well constrained, in particular in oceanic regions.

Likewise, constraining the modeled lithospheric stress with observations is challenging due to poor spatial coverage by the World Stress Map data (Zoback, 1992; Lithgow-Bertelloni and Guynn, 2004; Heidbach et al., 2008). An alternative way

documented in the literature is to compare the strain rate estimated from the modeled deviatoric stresses (Ghosh et al., 2008) with the Global Strain Rate Map (Kreemer et al., 2003). However, the lithospheric stress in plate interiors (i.e. far from the plate boundaries) is not well constrained with the Global Strain Rate Map. Hence, a gradually increasing coverage of the observed global stress field data serves as a motivation for studies attempting a global comparison of the observed and modeled stress field patterns, including our present study.

To date, two distinct approaches have been adopted to study the origin of the lithospheric stress, and each has given a relatively good fit to the observed stress field. On the one hand, Bird et al. (2008) have estimated the lithospheric stress from a model that disregards the mantle flow contribution and used the fit to the observed plate velocities as a sole criterion. On the other hand, Ghosh et al. (2013), Ghosh and Holt (2012), Lithgow-Bertelloni and Guynn (2004), Steinberger et al. (2001) and Wang et al. (2015) have aimed at assessing the influence of the mantle flow on the lithospheric stress field and have shown

that the bulk mantle flow explains a large part (about 80-90 %) of the stress field accumulated in the lithosphere (Steinberger et al., 2001), in both magnitude and the most compressive horizontal direction. The aim of the present study is to evaluate the contribution of the upper mantle above the transition zone to the observed spatial stress regimes of the lithosphere, while testing different approaches and data sets used to describe the thermal and rheological structure of the upper mantle. Building on earlier studies, we use a new global model of the lithosphere and mantle to compute the lithospheric stresses and topography.

We use a numerical method that allows for a separate treatment of the small-scale features in the upper mantle and large-scale



dynamic patterns in the lower mantle in a single calculation, with the upper boundary treated as a free surface (Sobolev, 2009). Deriving all force contributions from a single calculation resolves any inconsistency that might arise from treating individual force contributions to the stress field separately, as has been done in earlier studies (Bird et al., 2008; Steinberger et al., 2001; Lithgow-Bertelloni and Guynn, 2004; Ghosh et al., 2008; Naliboff et al., 2012; Ghosh et al., 2013; Wang et al., 2015).

In this study, a 3D global lithosphere-asthenosphere finite element model (Popov and Sobolev, 2008) with visco-elasto-plastic rheology is coupled to a spectral model of mantle flow (Hager and O'Connell, 1981) at 300 km depth. As part of this work, we estimate dynamic topography and correlate our results with two different residual topography models. One is based on seismic surveys of the ocean floor used to correct for shallow contributions to topography and free-air gravity anomalies on continents (Hoggard et al., 2016). The second model is taken from Steinberger (2016) and is based on actual topography

corrected for crustal thickness and density from CRUST1.0 (Laske et al., 2013). Both models are also corrected for subsidence of sea floor with age. To derive our stress model we have combined CRUST 1.0 with the thickness and thermal structure of the lithosphere estimated by Artemieva (2006) in continents, and a half-space cooling model of the ocean floor with age (Müller et al., 2008). This is an improvement compared to much simpler representations of the upper mantle structure in previous studies using a thin-sheet/shell approximation (Steinberger et al., 2001; Lithgow-Bertelloni and Guynn, 2004; Ghosh and Holt,

2012; Ghosh et al., 2013; Wang et al., 2015). We also account for the presence of slab locations and their corresponding impact on the upper mantle temperature based on Steinberger (2000). Stresses induced by regional and global variations in the crustal and lithospheric structures are of the order of 100 MPa in magnitude across strongly uplifted continental areas (Artyushkov, 1973). It is therefore clear that including a variable lithosphere and crustal thickness in calculations is preferable over the use of the thin-sheet/shell method (Steinberger et al., 2001; Bird et al., 2008; Ghosh and Holt, 2012). In addition we

have used a seismic velocity model to derive an alternative model of the lithosphere thickness and thermal density structure to counterpose the results from two simulations. While studying the impacts of the shallow (upper mantle) thermal and density anomalies and lateral variations in the rheological properties on the present-day dynamic topography and lithospheric stress state, we also attempt to quantify the uncertainties in the thermal structure of the upper mantle and their potential effects on the dynamic topography. As part of this analysis, we test the ability of our new 3D thermo-mechanical model of the lithosphere and

asthenosphere coupled to the mantle flow to reproduce the spatial pattern of the topographic anomalies and use it to separate out the effects of the chemical depletion in cratonic regions on the dynamic topography and lithospheric stress field.




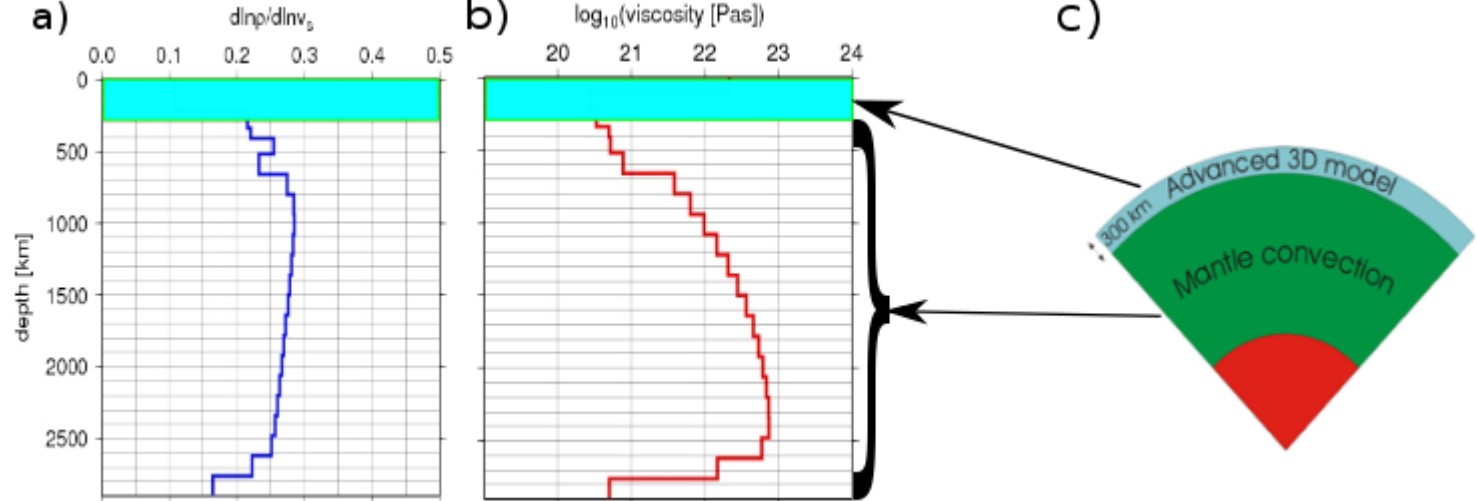

**Figure 1.** Adopted from Osei Tutu et al. (2017) (a) Depth-dependent scaling profile of S-wave velocity to density; (b) radial mantle viscosity structure (Steinberger and Calderwood, 2006) and (c) a schematic diagram of the numerical method that couples the 3D-lithosphere-asthenosphere code SLIM3D (Popov and Sobolev, 2008) to a lower mantle spectral flow code (Hager and O'Connell, 1981) at a depth of 300 km

## 2 Method

### 2.1 Model description

Our global numerical model of the Earth interior consists of the particle-in-cell finite element model SLIM3D (Popov and Sobolev, 2008) within the top 300 km, which solves coupled momentum and energy equations with a semi-Lagrangian Eulerian
grid and Winkler boundary condition. This allows for a free surface top boundary condition and a dynamic bottom boundary condition, achieved through coupling to a spectral mantle flow code (Hager and O'Connell, 1981) to account for the deep mantle contributions. There is no material exchange across the coupling interface, but the continuity of tractions and velocities is ensured through the Newton-Raphson iteration method. Figure 1(c) shows a sectional schematic representation of the coupled numerical model with depth-dependent layered mantle viscosity structure (Figure 1b) and seismic velocity-to-density scaling
profile of Steinberger and Calderwood (2006) (Figure 1a), which are only considered below the depth of 300 km. The top thermo-mechanical component (SLIM3D) has been used in a wide range of 2D and 3D regional numerical studies of crustal and lithospheric deformations (Popov and Sobolev, 2008; Brune et al., 2012, 2014, 2016; Popov et al., 2012; Quinteros and Sobolev, 2013) with different spatial and temporal resolutions but the coupled code is used here and in Osei Tutu et al. (2017) for the first time. In this 3D global study, we distinguish three material layers (phases) within the top component (SLIM3D):
the crustal layer, the lithosphere and the sub-lithospheric mantle layers in order to account for the stress and temperature-dependent rheology in the presence of major continental keels and the uppermost part of the subducted lithospheric plates. The



visco-elasto-plastic rheology is described in detail by (Popov and Sobolev, 2008), with specific modeling parameters given in Osei Tutu et al. (2017) and here in the appendix.

This study complements our previous study (Osei Tutu et al., 2017) about the influence of plastic yielding at plate boundaries on plate velocities in a no-net-rotation reference frame and on lithospheric net rotation. A forward model is run for half a million years with a time step of 5kyr, and at each time step tractions in the lower mantle due to density heterogeneities are computed using the spectral mantle code and then passed across the coupling dynamic boundary to the top component SLIM3D. Within the upper domain (SLIM3D), the flow velocities are then computed and passed back across the coupling boundary as an upper boundary condition to the spectral mantle code, with the method convergence estimated by comparing the velocity and traction norms of two successive iterations. Within the upper mantle, our crustal rheology is taken from Wilks (1990) and below the crust we have considered dry and wet olivine parameters in the lithosphere and sub-lithospheric mantle layers, respectively, modified after the axial compression experiments of Hirth and Kohlstedt (2004) (shown in the appendix, Table A1. Adopted from Osei Tutu et al. (2017) for studying the influence of both the driving and resisting forces that generate global plate velocities and lithospheric plate net rotation).

## 2.2 Thermal and density structures of the upper and lower mantle

We assign densities of the uppermost layers according to the crustal model CRUST1.0 (Laske et al., 2013). Underneath, we separately consider the layers below and above the interface between the two codes placed at a depth of 300 km to differentiate between the deep and shallow signals. Here the topographic signal induced by the layers below 300 km is assumed to be due to convection in the viscous mantle, although cold rigid subducting slabs (Zhong and Davies, 1999; Faccenna et al., 2007) and possibly also the deepest cratonic roots (Conrad and Lithgow-Bertelloni, 2006) extend deeper than 300 km. We use a 3D density structure inferred from the hybrid seismic tomography model of Becker and Boschi (2002) and apply a velocity-to-density conversion profile (Figure 1a) for the lower mantle buoyancy. In the upper mantle we test two different models for the representation of the upper mantle thermal and density structures, namely TM1 (Figure 2a) and TM2 (Figure 2b). TM1 is based on a 3D thermal structure inferred from sea floor age (Müller et al., 2008) for the mantle under oceanic regions. We use a half-space cooling model to infer the temperature $T_{ocean}$ as a function of age and depth according to:

$$\mathrm{T}_{ocean}(z,\tau) = \mathrm{T}_s + (\mathrm{T}_m - \mathrm{T}_s)\, erf\left(\frac{z}{2\sqrt{k\tau}}\right) \tag{1}$$

where $k = 8 \cdot 10^{-7}\mathrm{m^2 s^{-1}}$ is the thermal diffusivity, $\tau$ is the age of the oceanic lithosphere, $T_s$ is the reference surface temperature, $T_m$ is the reference mantle temperature, with $z$ being the depth beneath the Earth's surface. This is combined with the TC1 model across continents (Artemieva, 2006). In regions of continental shelf, where there is neither age grid nor heat flow data, we interpolated the resulting thermal structures surrounding these regions while in Iceland TC1 model was assigned.

The second model of the upper mantle thermal structure (TM2) is inferred from the seismic tomography model SL2013sv (Schaeffer and Lebedev, 2013). We have chosen this model because of its detailed representation of the upper mantle heterogeneities, which has been shown by Steinberger (2016) to allow a better prediction of the dynamic topography than previous models. This makes it a good candidate for comparison with the model results obtained using TM1, and for a regional investi-



**Figure 2.** The thermal structure of the upper mantle at a depth of 80 km from two reference thermal models adopted in this study. a) TM1, a heat flow-based thermal structure inferred from the TC1 model of Artemieva (2006) in the continents and the sea floor age model of Müller et al. (2008) in the oceanic areas. b) TM2, the thermal structure of the upper mantle inferred from the S-wave tomography model SL2013sv of Schaeffer and Lebedev (2013). The "ring-ing" visible in the upper structure is a side effect introduced by smoothing sharp boundaries with a spherical harmonic expansion.





gation of the upper mantle contribution to the lithospheric stresses and topography. Here we convert seismic velocity anomalies $\delta V_s$ into thermal anomalies $\Delta T$ within the upper mantle according to the relation:

$$\Delta T = \frac{\left(\frac{\delta V_s}{V_s(z)}\right)}{\left(\frac{\partial \ln V_s}{\partial T}\right)_P}, \tag{2}$$

where the subscript $P$ stands for a partial derivative at constant pressure (i.e. depth) based on Steinberger (2007). As a first

step, we do not correct for the effect of the chemical depletion in cratons in order to evaluate its influence on the modeled lithospheric stress field and topography. In addition and for comparison purposes, we introduce two other thermal models based on two different seismic tomography models SAW24B16 (Mégnin and Romanowicz, 2000) and S20RTS (Ritsema et al., 2011) to evaluate their performance relative to our reference seismic tomography model SL2013sv (Schaeffer and Lebedev, 2013) (Figure 2b). In our model setup, we define the reference crustal, lithospheric and asthenospheric densities (Table A1)

and account for lateral density variations linked to thermal anomalies (Figure 2) using the relation:

$$\rho(\Delta T) = \rho_{ref} \left[ 1 - \alpha \Delta T + \frac{P}{K} \right], \tag{3}$$

where $\rho_{ref}$ denotes the reference density at a reference temperature of 20°C and zero pressure, $\alpha$ denotes the thermal expansion coefficient chosen to be $2.7 \times 10^{-5} \mathrm{K}^{-1}$ in the crustal layer and $3 \times 10^{-5} \mathrm{K}^{-1}$ within the lithospheric and asthenospheric mantle and $K$ is the bulk modulus (Table A1).

## 3 Results and Discussion

### 3.1 Average creep viscosities and corresponding basal tractions

We start with estimates of global dynamic geoid, plate motion velocity and the resulting shear tractions at depth 300 km shown in Figure 3(a-c) to test our prescribed lateral viscosity variations within the upper mantle (see the slice in figure 3(d) for illustration) yielding realistic results. We compared our predicted geoid (Figure 3a) calculated with a 3-D viscosity

structure within the upper 300 km to the observation-based GRACE model (Reigber et al., 2004) (Supplementary Figure S2a) yielding a correlation of 0.85 at spherical harmonic degree 31. We also compared it to the geoid estimate from the simulation using a layered/radial viscosity structure (Steinberger and Calderwood, 2006) for all depths (Supplementary Figure S2c) resulting in a relatively low correlation of 0.82.

In figure 4(b), we show profiles of the estimated effective creep viscosity for continents and oceans within the upper mantle

and the crust (300 km). The corresponding diffusion and dislocation creep estimates are shown in figure 4(a) using olivine parameters modified after (Hirth and Kohlstedt, 2004) (Shown in the Appendix, Table A1. Figure 4(b) shows how a laterally averaged (dependent on depth only) asthenospheric viscosity decreases with increasing water content (i.e. 100, 500, 1000 $H/10^6 Si$). The viscosities are averaged separately across the continental and oceanic regions (Figure 4b, dashed versus solid lines). The average oceanic viscosity profiles give amplitudes lower than the respective average continental viscosity magnitude

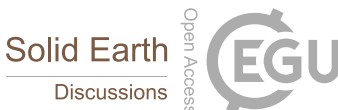





**Figure 3.** (a) Modeled geoid, (b) plate velocity and (c) horizontal and radial tractions at a 300-km depth calculated with LVV and TM1 thermal-density model in the upper 300 km and a 3-D density structure of the mantle inferred from Becker and Boschi (2002) in combination with the layered viscosity profile from Steinberger and Calderwood (2006) imposed below 300 km. (d) A slice of the TM1-based viscosity model at the latitude $25°S$ through the Nazca subduction plate.



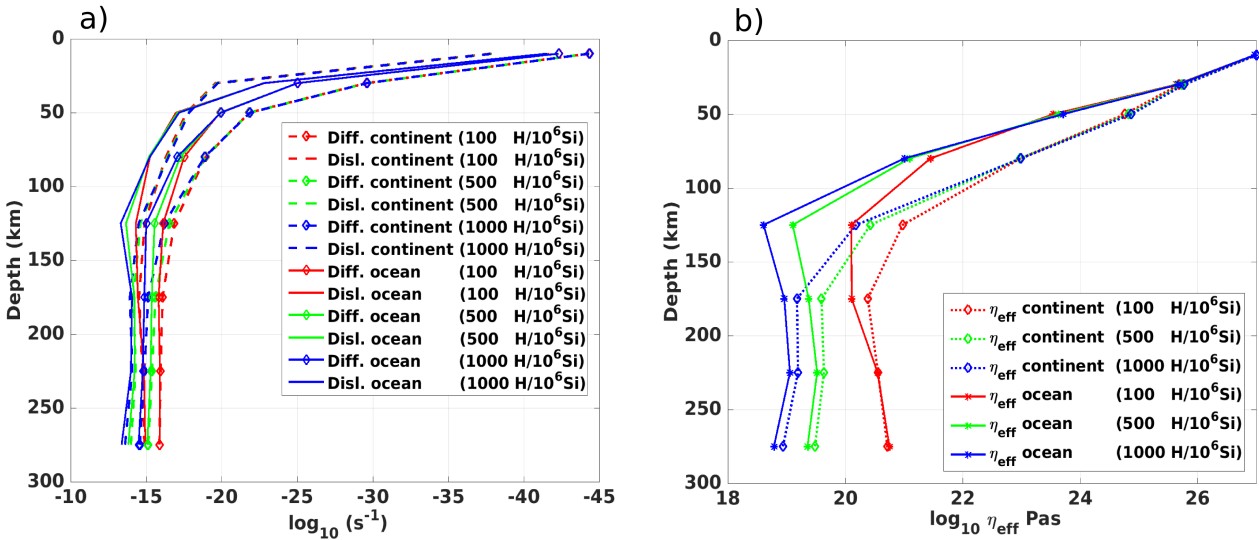

**Figure 4.** a) Calculated average strain rate versus depth for diffusion and dislocation creep across continents and oceans. b) The corresponding average creep viscosity versus depth in the upper mantle with olivine parameters.

($\eta_{eff}$), within the depth range of $100 \pm 60$ km. Seismological studies (e.g. Schaeffer and Lebedev, 2013; Kawakatsu et al., 2009; Fischer et al., 2010; Rychert et al., 2005) show this as a seismic wave velocity drop (~$5 - 10\%$), and as a transition between the lithosphere and asthenosphere corresponding to the low viscosity channel (Figure 4b). Figure 3(c) shows tractions causing stresses and topography in the lithosphere from the simulation using the creep parameters that correspond to the

green effective viscosity profile in figure 4(b), which was used for to model the dynamic geoid and plate motions. Here, at all plate boundaries we have used a friction coefficient $\mu = 0.02$ within the crust and lithospheric layers to generate the global plate velocities in a No-Net-rotation (NNR) reference frame shown in Figure 3(b) with RMS of 3.5 cm/yr. Since the focus of this study is to investigate the effect of the upper mantle lateral density variations on the horizontal stress field and dynamic topography, an assessment of the influence of the plate boundary friction and water content in the asthenosphere on

plate velocities has been carried out in a separate study (Osei Tutu et al., 2017). Hence, in the present work, we constrain our resulting creep viscosity with a cutoff for extreme viscosity values in the upper mantle by setting permissible minimum and maximum viscosity values similar to Becker (2006) and (Osei Tutu et al., 2017), with this approach yielding a good fit between the observed and modeled geoid.

## 3.2 Shallow and deep contributions to the crustal stress state

We start with examining separate contributions of the mantle heterogeneities below 300 km (deep Earth setup) and above (shallow Earth setup) to the global lithospheric stress field and topography. To calculate the contribution of the lower domain, we use a constant lithosphere thickness (100 km) and density (3.27 kg/m$^3$), a radial viscosity distribution shown in Figure 1(b) and a seismic velocity-to-density scaling (Figure 1a) below a depth of 300 km following Steinberger and



Calderwood (2006) and a 3D density structure below depth 300 km derived from the seismic tomography model Smean (Becker and Boschi, 2002). The resulting maximum horizontal magnitude ($SH_{max}$) and direction of the lithospheric stress field and the resulting dynamic topography are shown in Figures 5a and 5c. We have obtained compressional regimes in regions of past and present subduction. In the North and South American continents, beneath which the ancient Farallon and

Nazca plates were subducted, compressive stress magnitudes reach about 40 MPa. In the Far East, downwelling flows stretching from north to south from the northwestern Pacific through Australia towards Antarctica create compressional stress regimes with magnitudes ranging between ~ 50 and 80 MPa. These compressional regions are connecting the Arctic with Antarctica and engulf two distinct regions with extensional stress regimes centered on the Pacific and African superswell regions. The predicted $SH_{max}$ directions in Figure 5(a) generally follow the first-order lithospheric stress pattern (Zoback

1992), similar to previous mantle flow predictions of lithospheric stresses (Steinberger et al., 2001; Lithgow-Bertelloni and Guynn, 2004; Ghosh and Holt, 2012). In the largest extensional regions such as found in the Pacific superswell and above deep upwellings across southeastern Africa, stresses reach magnitudes of around 30 MPa. The modeled extensional/compressional pattern in the constant lithosphere, which get smoothed out over large distance are induced by the gradient in the tractions (Figure 3c) coming from the mantle flow.

To investigate the contribution of the upper domain (300 km) to the stress field, we calculate the magnitude and direction using model TM1 (Figure 2a) combined with the CRUST 1.0 model (Laske et al., 2013) and disregarding mantle density variations below 300 km (i.e. both horizontal and vertical tractions below the depth of 300 km are set to zero). Comparison of the lithosphere stress predictions from our shallow (Figure 5a) and deep (Figure 5b) Earth setups reveals notable differences in the model-based stress regimes, magnitudes and directions in continental regions. If stresses are generated by the upper

domain only, then almost all continental regions are characterized by an extensional regime, with the largest stress magnitudes found in areas of high topography and orogenic belts, such as the Tibet and Andes highlands. Our stress predictions from the shallow Earth setup with laterally varying crustal and lithospheric densities in Figure 5(b) show stress magnitudes and patterns similar to Naliboff et al. (2012). However, as opposed to the results of Naliboff et al. (2012) we predict high compressional stress magnitudes at continental margins, which may in part originate from a finer treatment of the crust and

our temperature-dependent creep viscosity. Also the high compressional stresses along the subduction margins in Figure 5(b) are likely induced by the slab models included in our setup. Figure 5(d) shows the resulting topography beneath air (free surface) coming from the combined effect of crustal and lithospheric densities in the top 300 km without the deep mantle flow contribution.

### 3.3 Total lithospheric stresses and topography

Next we compute the combined effect of both the lower mantle buoyancy and the upper mantle heterogeneities on the global $SH_{max}$ magnitude and direction for comparison with the separate contributions discussed above and with observations. Note that this is not a linear superposition of the separate contributions, because changes in the properties of the upper 300 km lead to changes in the topography and stress caused by density anomalies below 300 km depth. The resulting $SH_{max}$ direction and magnitude (Figure 6a) due to the combined contributions of the upper and lower mantle show compressional regimes in areas





**Figure 5.** a) Model-based maximum horizontal stress magnitude and most compressive stress directions [$SH_{max}$] following the convention with compression being positive, originating from mantle flow driven by density anomalies below 300 km. (b) Same for structure of the top 300 km of the upper mantle, computed with the CRUST 1.0 model and TM1. (c) and (d) depict the corresponding topography beneath air (free surface)



similar to Figure 5(a), while muting almost all strong extensional stresses predicted by our simulation with the shallow Earth setup in continents (Figure 5b). Also, the predicted $SH_{max}$ orientation generally follows the first-order lithospheric stress pattern (Zoback, 1992), similar to predictions based on only density anomalies below the 300-km depth (Figure 5a), with some regional deviations. The dominance of the contribution from below 300 km to the lithospheric stress field orientation

becomes apparent when looking at the similarities between the $SH_{max}$ directions in Figures 5(a) and 6(a) and dissimilarities with Figure 5(b), especially in continents. Nevertheless, the contribution from the upper 300 km to the predicted stress magnitude is evident in areas with large crustal thickness in continents, such as Tibet and the Andes. The regions where extensional regimes are predicted with only the contribution from below 300 km (Figure 5a) correspond well with the extensional stress regions in the combined model (Figure 6a).

The result is not very different, when we use the thermal density model TM2 for the total lithospheric stress field prediction. Both the predicted $SH_{max}$ magnitude and direction with TM1 (Figure 6a) and TM2 (Figure 5b) show notable similarities in oceans and continents, owing to the strong contributions from below 300 km which are similar for both models. They show relatively high compressional stress magnitudes in subduction or convergence regions such as the Mediterranean, south of the Tibetan Plateau, south of Alaska, and the northwest Pacific extending through the Sumatra subduction zone and underneath

the Australian and Antarctic plates. However, the $SH_{max}$ compressional signal underneath North America in Figure 6(a) is muted and that of the South American region turns into an extensional regime along the Andes (Figure 6) with the inclusion of the mantle above 300 km and the crust. Similar to Figure 5(a) both predictions with TM1 and TM2 (Figures 6a and 6b) show $SH_{max}$ extensional regimes corresponding to the regions of upwellings and/or volcanism. However, the model with TM2 generates a much higher extensional magnitude of ~60 MPa in the North Atlantic region around Iceland, and around the

Azores and Canary hotspots, compared to TM1. Stress magnitudes are more alike in the Southern Pacific Rise and around southern Africa. Differences are in part due to the detailed and well resolved upper mantle structures in the S-wave model used to derive TM2 (Schaeffer and Lebedev, 2013), as opposed to the upper mantle structure in TM1, which is based on the sea floor age in oceanic regions (Müller et al., 2008) and slab temperatures from Steinberger (2000). Also in regions where the coverage of heat flow data is poor (e.g. in South America and Antarctica, (Artemieva, 2006; Pollack et al., 1993)), TM2

(Figure 5b) may give better results. TM2 predicts compressional stress under Antarctica and along the subducting Nazca plate in South America induced by downwelling flow. In these regions there are barely any heat flow data and TM1 remains largely unconstrained. Both modeling setup with the combined effects from the crustal structure model, the upper mantle thermal-density structure (either TM or TM2) and deep mantle contributions give topography (Figure 6c-d) similar to actual topography. Comparing the similarities between Figure 6(c-d) to Figure 5(d) shows much of the Earth's topography comes

from density variations at shallow depths < 300 km.

### 3.4 Lithospheric stress and 'dynamic' topography without crustal effect

Following the above prediction of lithospheric stress field and topography, we repeated the two simulations to compute the $SH_{max}$ and topography, but this time without crustal thickness variations (Figure 7) to distinguish isostatic contributions form non-isostatic contributions. The resulting stress magnitude and orientation from TM1 (Figure 7c) and TM2 (Figure 7d)





**Figure 6.** Predictions of the $SH_{max}$ magnitude and direction from combined contributions due to lower mantle flow and upper mantle from a) TM1 with crust model and b) TM2 with crust model. The corresponding model topographies are is shown in c) and d) respectively.





without the crustal contribution are quite similar to the respective previous results shown in Figure 6 that include the crustal

contribution. Here, the resulting topographies with TM1 (Figure 7a) and TM2 (Figure 7b) show similar amplitudes due to the

sea floor cooling and thickening along the ridges in the Atlantic, Indian and Pacific Oceans, peaking above ~1.5 km. With

TM1, which explicitly contains subducted slabs, narrow, deep trenches are computed above subduction zones, such as in the

northwestern Pacific and at the west coast of South America. Also the negative topography in the Sumatra plate boundary is

reproduced well with the TM1 model reaching a value ~ -1.8 km. Based on tomography (model TM2) the computed

topographic lows are wider and less prominent.

Predicted topography with TM2 is higher in eastern Africa (2 to 2.5 km), and highly elevated regions are more extensive.

Figure 7(a) with TM1 (based on sea floor age) shows relatively low topography amplitudes in the northwest of the Pacific

plate around Hawaii and towards the Mariana trench compared to Figure 7(b) with TM2 (based on the S-wave model

SL2013sv) corresponding to a mean regional temperature difference of about ~200°C between TM1 and TM2 (Figure 2a-b).

The 'dynamic' topography with TM2 replicates nearly all island chains associated with hotspots in and around the African

plate, in the Pacific and along the Atlantic opening. In the North Atlantic, the positive topography (Icelandic swell) due to the

Iceland plume-lithosphere interaction (Rogozhina et al., 2016; Schiffer and Nielsen, 2016) is more pronounced in Figure 7(b)

with TM2 based on the tomography of Schaeffer and Lebedev (2013). Here the heights exceed 2 km as compared to

Figure 7(a) with TM1 based on the ocean floor ages of Müller et al. (2008), showing values slightly below 2 km. The high

isostatic topographic amplitudes along the mid-ocean ridges (MORs) as a result of high temperatures beneath these spreading

centers where new sea floor is created are generally more pronounced in the TM2 model simulation than in the TM1

experiment. Despite the striking differences between topographic amplitudes in Figures 7(a) and 7(b) along the MORs,

modeled stress orientations (Figure 7c-d) are very similar in these regions.

Also, the predicted strong negative topography (Figure 7a-b) in continental regions such as North America, Eurasia, western

Africa, South America, and western Australia are mostly due to mantle lithosphere in cratons. Low temperatures in the

thermal model TM2 (Figure 7b) are due to conversion from seismic models to temperature and density, based on the

assumption that all seismic velocity anomalies are due to thermal variations only. This produces unrealistically strong density

anomalies and hence, large negative topography in cratons (Forte and Perry, 2000), if correction due to the chemical depletion

in the mantle lithosphere is not considered. Cammarano et al. (2011) showed that correction for the depletion of the

lithosphere increases the inferred temperature of a cratonic root by about 100 K and decreases density by about 0.1 $gcm^{-3}$,

and fits observations well compared to models assuming pyrolitic composition. Hence we adopt two additional thermal

structures from different seismic tomography models SAW24B16 (Mégnin and Romanowicz, 2000) and S20RTS (Ritsema

et al., 2011) with corrections applied to the depleted mantle based on Cammarano et al. (2011) and compare with our results.

A follow-up estimate of stresses and topography using TM2 with a constant temperature of 100 K applied as a correction to

continental depletion in all major cratons is compared with observed fields. Also the large negative topography amplitude in

cratons observed in dynamic topography with TM2 compared to TM1 does not readily translate into similarly large variations

in the respective predicted $SH_{max}$ orientation (Figure 7c-d), showing that cratonic roots have less influence on the

lithospheric stress field (Naliboff et al., 2012).





**Figure 7.** Modeled 'dynamic' topography using the upper mantle structure (a) TM1 and (b) TM2 and corresponding $SH_{max}$ prediction with (c) TM1 and (d) TM2. In contrast to Figure 6, the effect of the crust is not included here.





## 3.5 Modeled versus observed lithospheric stress field

We compare our predicted $SH_{max}$ orientation to the observational stress data. Following the stress interpolation method presented by Müller et al. (2003), we used their Fixed Search Radius (FSR) method which uses a global weighting defined by a fixed Euclidean distance for the stress data interpolation and stress quality. The smoothed stress field orientation at a grid

point is based on the dominant stress data orientation within the selected radius. For a detailed explanation on the FSR method see Müller et al. (2003). Stress data with quality A, B, and C with known stress regime were considered. Since we do not consider the respective regime in our quantitative analysis, we also included the stress data with unknown style having quality A and B in our smoothing procedure to make our smooth field more robust. We smoothed the observed $SH_{max}$ orientation of the World Stress Map 2016 (WSM2016) (Heidbach et al., 2016), with a search radius of 270 km (Figure 8a-b) on a grid

interval of 2.5° x 2.5°. The background dot colors in the smoothed map represent the stress data regimes with red denoting normal fault, blue as thrust fault, green as strike-slip fault and black as unknown regime. For the interpolation we only took into account the orientation pattern of the stress data. We limit our comparison with modeled lithospheric stress orientation to areas with enough data for interpolation. The new WSM2016 has a relatively good coverage in some regions that were not well covered in the previous version (Heidbach et al., 2008) such as Brazil, parts of North America, Eastern Russia, and

Central Africa. We regard it as appropriate to compare the modeled stress orientation with the smoothed observational stress data and regard deviations of actual stress from smoothed stresses as a second-order pattern.

In Figure 8(c) we have superimposed our total modeled stress fields with the TM1 results depicted by thin bars and plotted on top of the TM2 results shown by thick bars. At long wavelength there is a fairly good agreement in the predicted stress orientation and regimes using TM1 and TM2. The smaller-scale contribution from the upper 300 km generates regional

variations in the stress pattern and regime in Figure 8(c) which are mainly due to density contrasts in the lithosphere or underneath (which are nearly isostatically compensated or cause lithosphere flexure) and due to topography (Zoback, 1992; Zoback and Mooney, 2003; Bird et al., 2006). Compared to the observed $SH_{max}$ patterns and regimes (Figures 8a and 8b) we predict similar styles in regions such as eastern Africa and Tibet with normal faulting comparable to earlier works that considered the effect of the whole mantle including lithosphere and crustal models (Lithgow-Bertelloni and Guynn, 2004;

Ghosh and Holt, 2012; Ghosh et al., 2013; Wang et al., 2015). We predict normal faulting mostly in regions above upwellings (mostly extensional regions) such as the Icelandic swell, Eastern African rift, or along divergent plate boundaries, while thrust faults are mainly predicted in compressional regions such as subduction zones and some other tectonically active regions in continents. In continental areas, few regional variations occur in South America, West Africa and on the Eurasian cratons. In oceans we see variations in the North Atlantic around the Icelandic swell, at the east Pacific Rise and around the southern

African plate region.

### 3.5.1 Angular misfit between WSM2016 and modeled lithospheric stress

To further evaluate the influence of each thermal structure we performed a quantitative comparison between modeled and smoothed observed stress orientations. The estimated angular misfit (Figure 9) is a measure of the minimum angle between



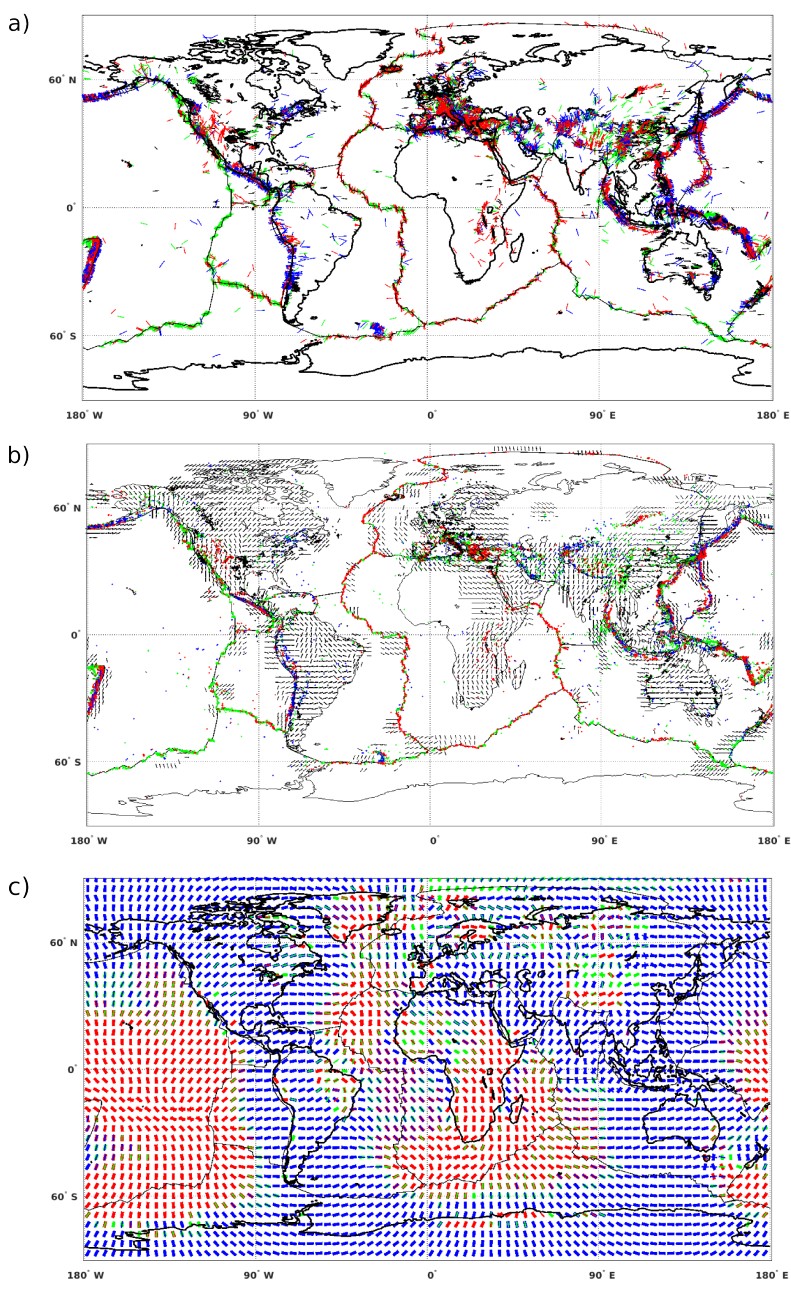

**Figure 8.** (a) World stress map 2016 (Heidbach et al., 2016), (b) interpolated World Stress Map data on a grid of $2.5°$x$2.5°$, using only stress orientation with a constant search radius 270 km, and (c) predicted $SH_{max}$ orientation and regime from total stress contribution with TM1 (plotted in thin bars) over TM2 (thick bars) upper mantle thermal structures. Colors of bars in (a) and (c) and dots in (b) indicate observed or predicted stress regime with red for normal faults or tensile stress, blue for thrust faults or compressive stress, and green for strike-slip faults or intermediate stress (one principal horizontal stress positive, one negative)



**Figure 9.** Angular misfit between the observed (WSM 2016) and total modeled stress directions with (a) TM1 and (b) TM2 upper mantle thermal and density structures.



the modeled lithospheric stress orientation (Figure 8c for TM1 and TM2) and smoothed observed stress orientation (Figure 8b), which ranges between 0° to 90°. Here, angular misfit lower than 22.5° is regarded as representation of a good agreement between the modeled and observed stress orientations, with values above 67.5° regarded as indicative of a poor fit. The general SSW to NNE stress orientation observed over the North American plate is matched by our model predictions

with both thermal structures TM1 and TM2. The angular misfit maps over North America obtained with both thermal structures show a poor fit over the Yellowstone and Rocky Mountains extending to the Great Plains (Ghosh et al., 2013). The observed localized NW to SE stress direction deviates (Figure 8a-b) from the predicted long-wavelength stress pattern (Ghosh et al., 2013; Humphreys and Coblentz, 2007). Even though the thermal model TM2 includes high-density cratonic roots, compared to TM1, their respective results for the angular misfits show that the North American cratonic root has a limited

influence on the stress field. The two density structures TM1 and TM2 yield mean values of 22.2° (std = 19.6°) and 22.9° (std = 20.7°), respectively. As the upper mantle thermal structure TM1 for the South American continent is not well-constrained, due to lack of heat flow data, the predicted stress field in continental Brazil gives a relatively poor fit, with a mean misfit of 37.73° (std =20.24°). However, TM2 does not perform much better resulting in a mean misfit of 33.79° (std = 21.9°). Both models fail to match the observed stress field in the Andes, where the dominant localized N-S orientation is predicted, mainly

as a results of the high topography and large crustal thickness (compare to Figure 7c-d). In the African continent, predicted N-S stress orientations along the Eastern African Rift from either model match the observed stress quite well with TM1 fitting observations much better compared to TM2, but both fail over the Congo craton and the South African plateau.

### 3.5.2 Western Europe

The stress field in western Europe is influenced by the North Atlantic ridge (NAR) push in the west and possibly by the

far-field slab pull from the north-western Pacific subduction zones, while in the south, the driving forces are induced by the convergence of the African and Eurasian plates, with Africa subducting under Eurasia in the Mediterranean (Zoback, 1992; Müller et al., 1992; Gölke, 1996; Heidbach and Höhne, 2007; Schiffer and Nielsen, 2016). These plate boundary forces combined with the anomalous mantle pressure (Schiffer and Nielsen, 2016) underneath the North Atlantic lithosphere generate the dominant first-order NW-SE stress pattern. In our study, due to mantle contribution >300 km, we could match the

NW-SE stress orientation nearly perfectly, with the model using TM1 (Figure 10a) showing small regional deviations, while the use of TM2 (Figure 10b) results in larger deviations from this NW-SE pattern. It gives a second-order stress pattern in some regions, such as Poland, Hungary, Romania, Turkey, Russia and France.

These regional pattern deviations between modeled and observation orientations are mainly induced by differences in the upper mantle density structures and topographies (Heidbach and Höhne, 2007) (compare to Figure 5a). The high density of

heat flow data (Pollack et al., 1993; Artemieva, 2006) in continental Western Europe (TM1) improves the fit to the observed stress field compared to the thermal structure based on S-wave velocity (TM2) yielding mean misfit values of 18.30° (std = 22.67°) and 19.9° (std = 22.64°), respectively. None of our models was able to predict the E-W stress orientation in the Aegean-Anatolian region coming from the ongoing subduction slab rollback (Heidbach and Höhne, 2007; Heidbach, 2003), giving angular misfits greater than 40° (Figure 10a and Figure 10b). The large angular misfit obtained in this region could





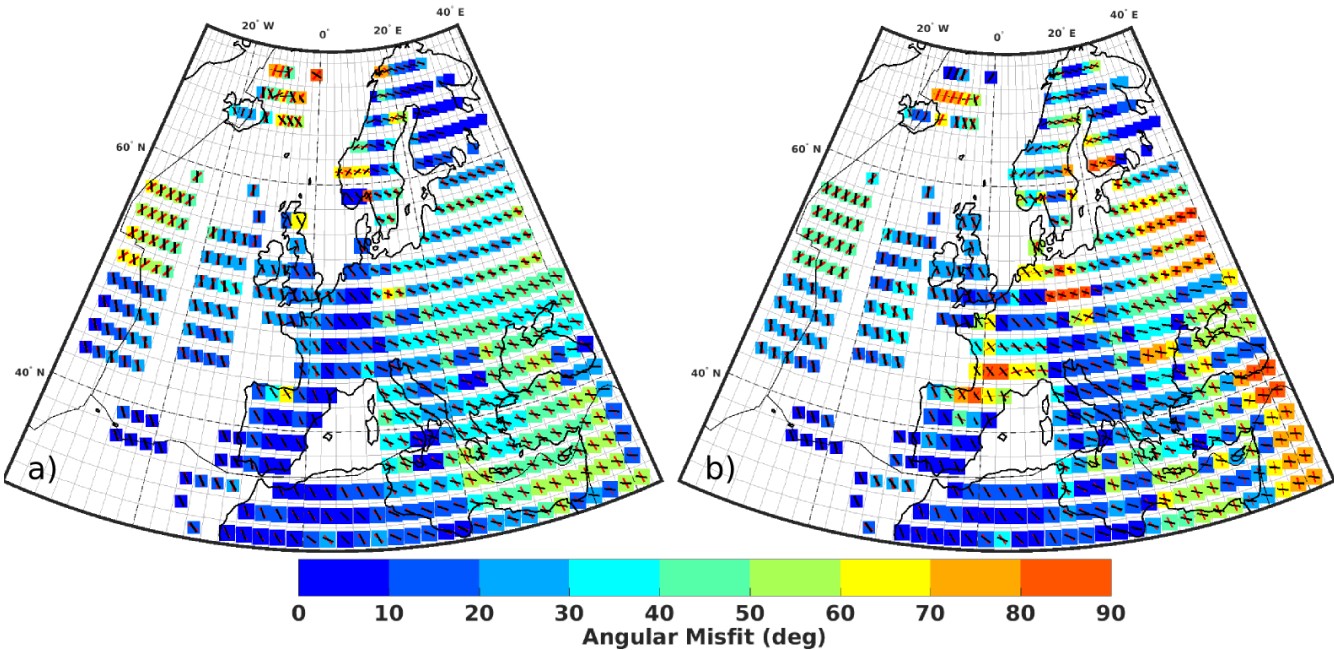

**Figure 10.** Angular misfit in Europe between the observed and modeled total stresses with (a) TM1 and (b) TM2. Red bars denote modeled orientations versus black bars showing the smoothed observed stress field (WSM2016).

stem from neither of the two upper mantle thermal-density structures (TM1 and TM2) used is able to accurately capture the dynamics in the Hellenic/Aegean arc and the subducted plate, hence a regional treatment will be appropriate (Faccenna et al., 2014).

### 3.5.3 Tibet and surrounding regions and Australia

5  The Australian continent has a similarly large amount of heat flow data. Hence, the predicted intra-plate stress pattern with TM1 results in a somewhat lower angular misfit (Figure 11a) with mean value 23.07° (std = 19.4°) than with TM2 (Figure 11b) (mean = 32.7° and std = 24.22°). It has been argued that the stress pattern in Australia is mainly driven by plate boundary forces (Reynolds et al., 2002), but based on the lithospheric and crustal structures used we show here that crustal and sub-lithospheric heterogeneities have a certain degree of influence. Nonetheless the W-E stress direction in the western

10  provinces and NE-SW direction in the north are matched by either model quite well. The orientation of the modeled and observed (smoothed) stress patterns deviates from the absolute plate velocity direction by almost 90° in the entire plate unlike North America. The collision of India and Eurasia forming the Himalayas results in a complex crustal and lithospheric deformation (van Hinsbergen et al., 2011; Gaina et al., 2015) maintaining the predicted NE-SW compressional stress. The $SH_{max}$ predictions with TM1 (Figure 11c) fit better the stress pattern over the Tibetan Plateau with a mean value of 28° (std



**Figure 11.** Angular misfit in Australia (a and b) and in and around Tibet (c and d) between the observed stress and modeled total stress with TM1 (a and c) and TM2 (b and d) upper mantle structures. Red bars denote modeled orientations versus black bars showing the observed stress field (WSM2016)





= 23°) compared to TM2, where a predicted E-W direction results in a misfit ~50° (Figure 11d). Both models performed relatively poorly over parts of China, when compared to the observed stress field.

### 3.6    Comparing the modeled dynamic topography to the observed residual topography.

Here, we compare our modeled dynamic topography to two independent observation-based residual topography fields

(Hoggard et al., 2016; Steinberger, 2016). Residual topography gives a convenient way to constrain both isostatic and non-isostatic contributions to the modeled dynamic topography (Crough, 1978; Gurnis et al., 2000; Wheeler and White, 2000; Becker et al., 2014; Heidbach et al., 2016; Steinberger, 2016). This is done with the assumption that if topography is perfectly compensated isostatically within the upper mantle at depths within the range of 100 - 150 km, the integral of density with depth, as a function of crustal thickness and density to the Moho depth and of sea floor age will be the same everywhere for

the chosen depth. The observation-based model by Hoggard et al. (2016) is derived from ocean seismic surveys (in-situ) in oceanic regions and free-air gravity anomaly data in continents (Figure 12a), while the residual topography model of Steinberger (2016) (Figure 12b) is derived with the CRUST 1.0 model (Laske et al., 2013). These two models are comparable in most oceanic regions, but give large mismatches in continents. For example, the subducting plate under South America induces a negative anomaly in Figure 12(b) but in the same region there is a positive anomaly in Figure 12(a) due to the

free-air gravity data used across continents. Hence, we perform a regional quantitative comparison for oceans and continents separately. To compare the modeled dynamic topography from TM1 and TM2 simulations (Figure 7a and 7b) to the observed fields (Figures 12a and 12b), we first remove the height due to ocean floor cooling. This is done by subtracting the height estimates from sea floor age (Müller et al., 2008) from the modeled dynamic topography, using the relation $H_{topo} = 3300m \cdot (1 - \sqrt{\frac{age}{100Ma}})$. Here we assume a half-space cooling for the sea floor with age. For a smooth transition of

topographic height from ocean to continent and to avoid large jumps we nominally assume a 200 Ma lithosphere age for continents following the approach of Steinberger (2016). The resulting modeled dynamic topography fields (Figure 12c-d) with the effect of the sea floor cooling with age removed, and with locations of active hotspot volcanism plotted as green dots show to which extent each of the models is able to predict the positive topographic amplitudes due to upwellings induced by plume heads pushing the lithospheric base.

A qualitative comparison of the two observation-based residual topography fields with the modeled topography shows some features that are well reproduced with both the TM1 and TM2 models. Among them are the Pacific Swell and the Hawaiian plume track, while the Canary Island plume, and the heights around south-eastern Africa are much better reproduced by the TM2-based dynamic topography. Removing the height due to ocean floor age results in either zero or negative topographic amplitudes along MORs in the Atlantic and Indian Oceans in the TM1-based dynamic topography (Figure 12c), giving

correlation of 0.323 and 0.198 (Table 1) in oceans to Steinberger (2016)(S2016) and Hoggard et al. (2016)(H2016), respectively. This model uses the thermal density structure derived from the ocean floor age in the upper 300 km; hence, when this contribution is removed, only the lower mantle contribution remains. In contrast, the TM2 model still gives small-scale topography anomalies due to density anomalies other than from the sea floor cooling at depths above (300 km), which are resolved by the seismic model used to derive TM2 thereby giving relatively higher correlation to S2016 and H2016 of 0.348





**Figure 12.** Comparing a) the in-situ observed residual topography from Hoggard et al. (2016), and b) the residual topography based on the CRUST 1.0 from Steinberger (2016) with modeled dynamic topography using TM1 (c) and TM2 (d) upper mantle thermal density structures with the effect of the sea floor cooling with age removed. Green dots with black circles around show locations of major hotspots

and 0.284 in oceans respectively. To estimate the separate regional ratio between the modeled and observation-based residual topographies for continents and oceans, we assigned the mean continental amplitude in continental areas to estimate the degree by degree ratio for oceans only (Figure 13b) and vice-versa for oceanic regions to estimate continents ratio (Figure 13a).

5     In continents, the TM1 model (Figure 12c) is similar to the residual models (Figure 12b), exhibiting a correlation of 0.481 and a ratio of 0.98 (Figure 13a) up to the spherical harmonic degree 30. Over North America, Eurasia, and Australia it also fits the observed stress field better than TM2 (Figure 9). The TM2 model gives similar ratio and correlation, but at degrees lower than 15 the TM2-induced modeled dynamic topography is about twice the amplitude of TM1 (Figure 13a). Over the African continent with far less heat flow data used to derive TM1, the thermal density structure gives a large continental uplift up to

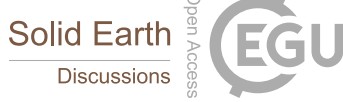

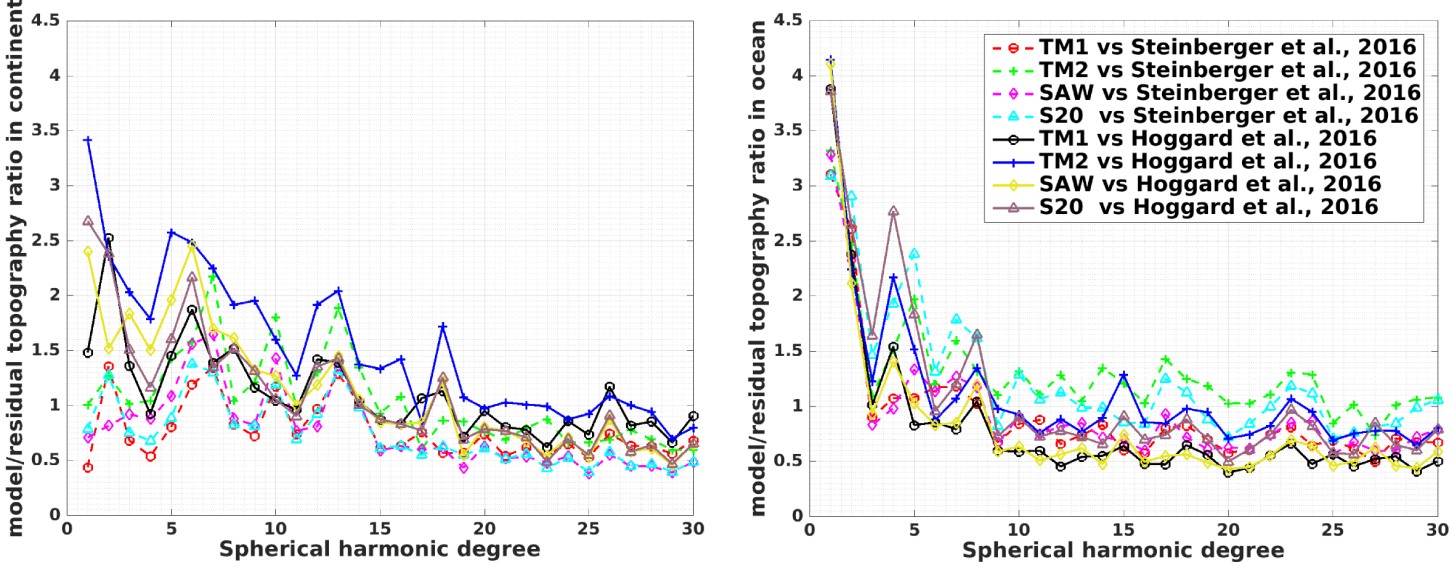

**Figure 13.** Ratio of modeled dynamic topography from TM1 and TM2 for (left) continental and (right) oceanic regions with observation-based residual topography from Steinberger (2016) and Hoggard et al. (2016)

about 2 km, similar to parts of Antarctica (Figure 12c). In Figure 12(d) this uplift is less extended, better resolving the negative topography of the Congo craton but reaching a height above 2 km over the East African swell similar to S2016 (Figure 12b). Many of the remaining continental regions, however, show large negative topographic magnitudes of -2 km and more, resulting from neglecting the compositional effects in cratons (e.g. Eurasia, Australia and North America). The wide

range variations shown in degree 1 to 2 ratio for continents Figure 13a) are due to the strong contributions coming from the different cratonic structures in each thermal model. To assess the robustness of our results we introduce two other upper mantle thermal density structures derived from SAW24B16 (Mégnin and Romanowicz, 2000) and S20RTS (Ritsema et al., 2011) seismic tomography models as a qualitative check for our TM2 model. For the seismic velocity anomalies from SAW24B16 (SAW) and S20RTS (S20) the method of Cammarano et al. (2011) is used to convert seismic tomography models to temperature structures taking into account chemical depletion in cratonic areas. In contrast, for TM2, we have assumed

additional 100 K converted to a negative density as compositional contribution in all cratons to the depth of 100 km as opposed to the more realistic treatment of compositional effects as done for SAW and S20. The modeled topography shows improvements in cratonic regions but there is almost no change in the resulting lithospheric stress field (Supplementary figure S3). The correlation to S2016 increases to 0.512 for TM2 (with an assumed 100 K compositional effect) in continents. SAW

and S20 give much higher correlation 0.653 and 0.718 in continents (Table 1), which could be a result of a more realistic treatment of cratonic regions but also of using different seismic tomography models. For example, Steinberger (2016) used a similar simple procedure to convert seismic velocities from different tomography models to density and still obtained a rather high correlation of 0.64 in continents.



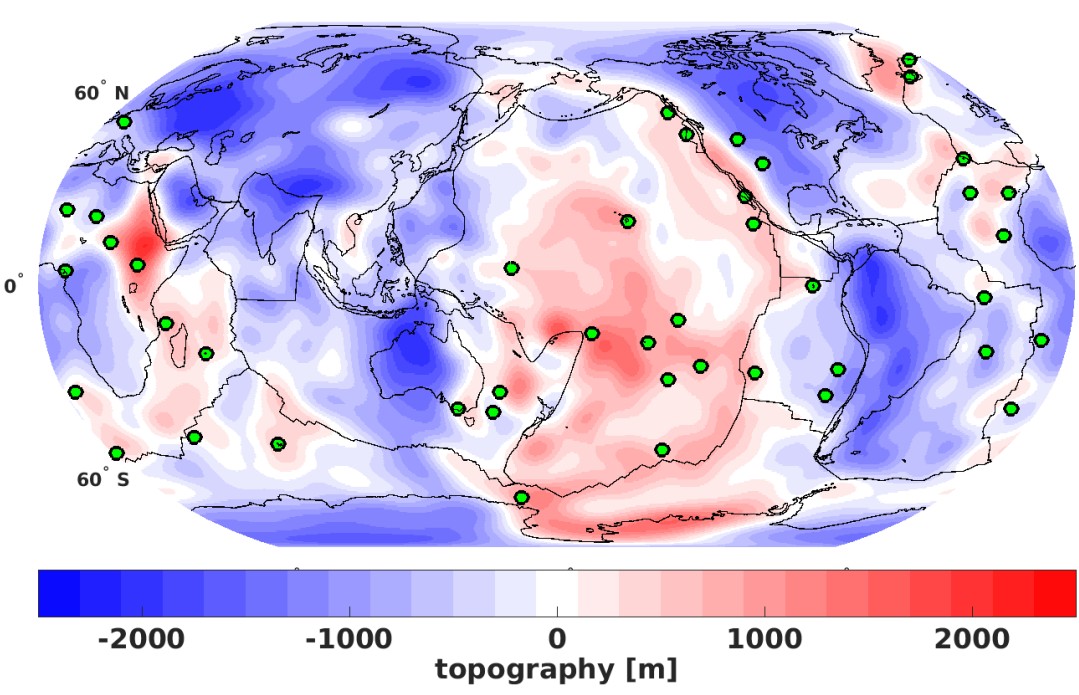

**Figure 14.** Modeled dynamic topography using TM2 upper mantle thermal density structures with constant temperature (100 K) added in cratons. The effect of the sea floor cooling with age is removed and green dots with black circles around showing locations of major hotspots.



**Table 1.** Correlation the between modeled dynamic topography and the observation-based residual topography models (Steinberger, 2016; Hoggard et al., 2016) for continents and oceans.

| *modeled topography* | Steinberger, 2016 | | Hoggard, 2016 | |
|---|---|---|---|---|
| *Upper mantle Thermal density* | *Ocean* | *Continent* | *Ocean* | *Continent* |
| $1. TM1$ | 0.323 | 0.481 | 0.198 | 0.169 |
| $2. TM2$ | 0.348 | 0.498 | 0.284 | 0.171 |
| $3. TM2 + 100K\,(in\,cratons)$ | 0.370 | 0.512 | 0.284 | 0.180 |
| $4. S20RTS\,(S20)$ | 0.442 | 0.653 | 0.221 | 0.232 |
| $5. SAW24B16\,(SAW)$ | 0.248 | 0.718 | 0.287 | 0.188 |

The assumed compositional correction is not very large giving about a 100 m reduction in the cratonic negative anomaly (Figure 14) compared to the case without correction in continents (Figure 12d). This in part supports the proposed treatment of the upper mantle thermal density structure with joint petrological and seismological constraints (Forte and Perry, 2000; Forte et al., 2010; Cammarano et al., 2011), which is outside the scope of our studies. The residual topography of Hoggard
et al. (2016) shows positive amplitudes over the Eurasian craton due to the free-air gravity data used, while the other residual (Figure 12b) and all modeled dynamic topography models give negative values, resulting in a low correlation with H2016 on continents for all models. Here the correlation with H2016 in oceanic regions is also lower except for SAW24B16. This result is model-dependent, and Steinberger et al. (2017) also find an improved correlation with H2016 in oceans using a different density model.

**4 Conclusions**

The aim of our study is to identify and quantify the influence of density anomalies and rheology in the crust and mantle on the present-day lithospheric stress field and dynamic topography. The focus is on anomalies and rheology above 300 km depth; therefore we use a number of different density structures, and nonlinear temperature and stress dependent rheology above 300 km: Our TM1 model is based on heat flow data on continents (Artemieva, 2006) and sea floor age (Müller et al., 2008) in
the oceans, while model TM2, and several alternative models considered, are based on seismic tomography (Schaeffer and Lebedev, 2013; Ritsema et al., 2011; Mégnin and Romanowicz, 2000). In contrast, only one density structure, based on the SMEAN (Becker and Boschi, 2002) tomography, and a radial viscosity structure (Steinberger and Calderwood, 2006) is used below 300 km depth. A key feature that distinguishes our work from previous studies is the use of a coupled code (Sobolev, 2009) that considers density heterogenety in the entire mantle, along with a realistic lithosphere with free surface, such that
lithosphere stresses are computed with a fully three-dimensional, rather than a thin-sheet approach.





Resulting lithosphere stresses are rather similar, both among the different models we consider, and to previously published results. They are also similar to a case where only the contribution from the mantle below 300 km is considered, showing that a larger portion of the contribution to the lithospheric stress field originates from mantle flow driven by density anomalies below 300 km depth (Steinberger et al., 2001; Lithgow-Bertelloni and Guynn, 2004). Only in some regions, particularly those with

large and variable crustal thickness, such as Tibet, or the Altiplano, shallow contributions are dominant. The lower mantle stress contribution is dominated by very-large-scale structures, with stress directions remaining similar over thousands of kilometers. It is related to very-large scale mantle structures, which are well imaged by seismic tomography, causing overall similarity between our models and published ones.

We compare computed directions of maximum compressive stress with the World Stress Map, and find a rather good overall

agreement, confirming previous comparisons. However, regional comparison highlights those areas where the fit remains poor: These include the Colorado Plateau, the Altiplano, parts of Brazil, the Congo Craton, and parts of China, thus highlighting regions on which future studies could focus. Computed stresses based on heat flow (Model TM1) compare more favorably to observations in those regions where heat flow coverage is good (e.g. Western Europe), whereas the stresses computed from tomography (Model TM2) give a better fit for regions of poor heat flow coverage, such as South America.

In contrast to stress field, density anomalies above 300 km depth contribute dominantly to dynamic topography. Therefore, dynamic topography is more variable among the different models we consider and differs more strongly from published models. Dynamic topography also has a larger contribution at smaller scales. Some of these contributions can be related to subducted slabs or mantle plumes. Confirming previous results, we find that negative topography in cratons is too large, unless a correction for the depletion of cratonic lithosphere is considered. The best fit can be obtained, if the method of Cammarano et al. (2011)

is used to convert seismic tomography models to temperature structures, taking chemical depletion in cratonic areas into account. The best agreement is found with residual topography on continents that considers crustal thickness variations based on CRUST1.0 (Laske et al., 2013) rather than deriving it from the gravity field. In order to fit either observable – stress or topography – attention has to be mostly paid to a detailed treatment of the Earth's parts – deeper or shallower – that give the largest contribution.





## Appendix A: Rheology of the upper mantle and lithosphere

The coupling between the lithosphere and the mantle in our model allows for an implementation of realistic rheological parameters in both model domains. In SLIM3D, the stress- and temperature-dependent rheology is implemented according to an additive strain rate decomposition into the viscous, elastic and plastic components:

$$\dot{\varepsilon}_{ij} = \dot{\varepsilon}_{ij}^{vis} + \dot{\varepsilon}_{ij}^{el} + \dot{\varepsilon}_{ij}^{pl} = \frac{1}{2\eta_{eff}}\tau_{ij} + \frac{1}{2G}\hat{\tau}_{ij} + \dot{\gamma}\frac{\partial Q}{\partial \tau_{ij}} \tag{A1}$$

where $G$ denotes the elastic shear modulus, $Q = \tau_{II}$ is the plastic potential function, $\hat{\tau}_{ij}$ is the objective stress rate, $\dot{\gamma}$ denotes the plastic multiplier, $\tau_{ij} = \sigma_{ij} + P\delta_{ij}$ is the Cauchy stress deviator, $P = -\sigma_{ii}/3$ is the pressure, $\tau_{II} = (\tau_{ij}\tau_{ij})^{1/2}$ stands for the effective deviatoric stress, and $\eta_{eff}$ is effective creep viscosity derived by combining the diffusion and dislocation creep mechanisms, as follows:

$$\eta_{eff} = \frac{1}{2}\tau_{II}\left(\dot{\varepsilon}_{diff} + \dot{\varepsilon}_{disl}\right)^{-1} \tag{A2}$$

The effective scalar creep strain rates are given by (Kameyama et al., 1999):

$$\dot{\varepsilon}_{diff} = A_{diff}d^{-p}\left(C_{H_2O}\right)^{r_{diff}}\tau_{II}\left(\frac{E_{diff} + PV_{diff}}{RT}\right) \tag{A3}$$

$$\dot{\varepsilon}_{disl} = A_{disl}\left(C_{H_2O}\right)^{r_{disl}}\left(\tau_{II}\right)^{n}\left(\frac{E_{disl} + PV_{disl}}{RT}\right) \tag{A4}$$

where the symbols $A$, $E$ and $V$ denote the experimentally prescribed pre-exponential factor, the activation energy and the activation volume, respectively, $R$ denotes the gas constant, $T$ is the temperature, $n$ is the power law exponent, $d$ is the grain size, and $p$ is the grain size exponent, $C_{H_2O}$ is water content in ppm H/Si, and $r_{diff}$ and $r_{disl}$ are the water content exponents.

Along plate boundaries we account for the brittle deformation, with the yield stress defined according to the Drucker-Prager criterion based on the dynamic pressure:

$$\tau_{yield} = c + \mu P \tag{A5}$$

where $c$ is the cohesion, $\mu$ is the coefficient of friction. Following (Sobolev, 2009) we use reduced friction coefficient values at the predefined plate boundaries (Bird, 2003) treated as narrow zones in the crustal and lithospheric layer in the depth range 0-80 km, and high friction coefficient of 0.6 in all lithospheric materials outside of the plate boundaries.





**Table A1.** The upper mantle creep viscosity is calculated using olivine parameters from the axial compression experiments of Hirth and Kohlstedt (2004). Crustal rheology is taken from (Wilks, 1990). The rheological parameters used in this study with varying Olivine water content of 100, 500, 1000 p.p.m $H/10^6 Si$ in the weak asthenospheric mantle with dry lithosphere material. For more details regarding the formulation of the physical model and numerical implementation the reader is referred to Popov and Sobolev (2008)

| Parameter | Unit | Crust | Lithosphere (strong mantle) | Asthenosphere (weak mantle) |
|---|---|---|---|---|
| $Bulk\,modulus\,K$ | GPa | 6.3 | 12.2 | 12.2 |
| $Shear\,modulus\,G$ | GPa | 4.0 | 7.40 | 7.40 |
| $Density\,\rho$ | $gcm^{-3}$ | 2.85 | 3.27 | 3.30 |
| $Cohesion\,c$ | MPa | 5.0 | 5.0 | 5.0 |
| $Friction\,coefficient\,\mu$ | - | 0.6* | 0.6* | 0.6* |
| Diffusion creep parameters ($d = 10$ mm $\quad p = 3 \quad r_{diff} = 1$) | | | | |
| $A_{diff}$ | $Pa^{-1}s^{-1}$ | - | $10^{-8.65}$ | $10^{-8.82}$ |
| $Activation\,Energy\,E_{diff}$ | KJ/mol | - | 375 | 335 |
| $Activation\,Volume\,V_{diff}$ | $cm^{-3}/mol$ | - | 6.0 | 4.0 |
| Dislocation creep parameters Dislocation ($r_{disl} = 1.2$) | | | | |
| $A_{diff}$ | $Pa^{-n}s^{-1}$ | $10^{-21.05}$ | $10^{-15.19}$ | $10^{-14.67}$ |
| $Activation\,Energy\,E_{diff}$ | KJ/mol | 445 | 530 | 480 |
| $Activation\,Volume\,V_{diff}$ | $cm^{-3}/mol$ | 10.0 | 17.0 | 14.0 |
| $Power\,law\,exponent\,n$ | - | 4.2 | 3.5 | 3.5 |





*Code availability.* The coupled global numerical code used to generate the results in this study builds on an in-house SLIM3D code (Popov and Sobolev, 2008) and spectral code of (Hager and O'Connell, 1981) and is available upon request from any of the authors

*Competing interests.* The authors declare that they have no conflict of interest.

*Acknowledgements.* This work was supported by GeoSim grants under the Geo.X program in conjunction with GFZ-Potsdam, University of
5  Potsdam and Free University of Berlin.





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
