# Peer review of "Effects of upper mantle heterogeneities on lithospheric stress field and dynamic topography"

_Solid Earth, 2017_

## Referee Comment (RC1) · Anonymous Referee #1 · 20 Nov 2017

This paper presents a new mantle flow model that couples global mantle flow (driven by density heterogeneity inferred from seismic tomography) to a detailed surface model for the upper 300 km of the mantle – this includes more detailed heterogeneity associated with upper mantle viscosity variations and density heterogeneity. The authors use this coupled model to predict both lithospheric stresses and dynamic topography, which are both the result of convective stresses from viscous flow in the mantle. The authors use 2 different predictive flow models (TM1 based on heat flow and seafloor ages and TM2 based upper mantle tomography model) and compare them to the global stress map for stresses and 2 different predictions of dynamic topography. They find that both predictive flow models (which differ only in the upper 300 km) predict similar stress fields but TM2 gives a better prediction of dynamic topography. The authors conclude

that dynamic topography is more sensitive to upper mantle structure than lithospheric stresses, that their model does a reasonably good job of predicting the lithospheric stress field, and that dynamic topography is better-predicted using seismic tomography data instead of heat flow and seafloor ages.

I think that the content of the paper useful and innovative – the new models are innovative and are – to my knowledge – the first of their kind to look at the impact of detailed upper mantle heterogeneity on stresses and dynamic topography. The predictions they make are useful for understanding the problem, and should lead to greater understanding of the system a. Thus, I think that the paper is eventually publishable.

The paper has a lot of information and is a bit hard to follow because the authors are comparing 2 predictive models (TM1 and TM2) with 2 output fields (stresses and dynamic topography), one of which has 2 different options (dynamic topography from Steinberger 2016 or Hoggard 2016). Thus, there are several combinations of comparisons, and it is a bit of an effort to follow what is being compared with what. On top of that, the writing is a bit wordy with some repetition. This makes reading and understanding the paper a bit laborious.

I think for this paper to be effective, it needs better organization and structure, and the writing needs to be simplified and shortened. I give some suggestions for this below. Given this, I recommend significant re-revision of the figures and text, although I do not think that significant new analysis is necessary. Thus, I would probably call for "moderate revision". Below is a an overall assessment of the changes that I think would be useful for improving the paper and also some specific points about the paper.

Major Points:

Overall Structure: One problem is that the predictions and analysis of stresses and dynamic topography are intermingled in section 3. I recommend separating this into two sections, one that is devoted to stresses and the other to dynamic topography. Thus, I recommend removing the topography predictions from Figs. 5 and 6 (so these

figures are devoted to stresses), and removing the stresses from Fig. 7 (so this figure is for dynamic topography). I also recommend revising the text around these figures to focus specifically on stresses and then on dynamic topography.

Introduction – I think the introduction is a bit too long and rambling, and should get to the main point more quickly – that this paper presents a new method for evaluating the role of upper mantle heterogeneity (both density heterogeneity and viscosity heterogeneity) for mantle stresses. There is much background – condensing it to just the relevant parts will help the authors emphasize their new contribution a bit more clearly.

Conclusions – I think that the paper does a nice job of concluding the major results. But there is no discussion section – it would be helpful to have a short discussion of the implications of this work for our understanding of dynamic topography and stresses, and other factors that might be important to include in the models in the future.

Figures: There are more figures than are necessary. In particular:

Figure 1 and 3 are illustrative about the model, but the information in these figures is not used much – perhaps only the essential components of Fig. 3 could be incorporated into Fig.1? (for example, the geoid, plate velocity, and tractions are not really used here)

Figure 4 shows comparisons for different models of dislocation and diffusion creep, and different water contents – however it is not clear to me that all of this complexity is necessary? Why not simplify by showing only the most relevant models and say in the text what is the impact of changing the rheological parameters?

As I mentioned above, I think that figures 5cd, 6cd, and 7cd are not necessary. Fig. 7 should be moved later into the dynamic topography section.

For the stress comparisons, I mention below that Fig. 8a is not necessary. Also Figures 10 and 11 could probably be combined.

Figures 7ab and 12cd both show dynamic topography for models TM1 and TM2, the

only difference being that the effect of seafloor age is removed from 12cd. I think that it is appropriate to only use Figs. 12cd and omit 7ab, because the models 12cd are compared to the "observation" models 12ab.

Figure 13 – I am a little unsure how the amplitudes of spherical harmonic degrees are compared for only parts of the globe (continents or oceans). The spherical harmonics are globally-defined, so does it make sense to compare the spherical harmonic components over the oceans-only or continents-only? How is the rest of the world considered when the spherical harmonic components of the rest of the world are defined? I wonder if it might be better to just do the comparison for the whole globe and avoid this problem?

Figure 14 – perhaps could be added as an alternative predictive model to Fig. 12?

Specific points:

Title – I think the word "the" should be added before "lithospheric stress field"

Page 1 – the statements on lines 7-8 and 9-11 basically say the same thing – I think this sort of repetition is not needed in the abstract

Page 1, line 12 – stating that a correlation has a value of 0.51 doesn't mean too much to a reader – is this correlation good or poor?

P1, L13 – it is not the lithospheric stresses that are being improved, it is the model fit to them.

P1, L15 – the difference in angular misfits reported here (18.3 vs 19.9 deg) doesn't seem very significant – I'm not sure that it is useful to include this information

Abstract – generally, it would be good to close the abstract with an overall general statement about what the reader should take away from this study. Currently the abstract doesn't do this.

P2, L 27 – this paragraph starts "At longer wavelengths," which seems to differentiate

it from the previous paragraph. But both paragraphs are about long wavelengths. It would be better to distinguish this paragraph by stating out the onset that it deals with dynamic topography.

P3, L 27 – Here the Bird (2003) approach is described as using the fit to the observed plate motions – this needs a clearer explanation because it contradicts the statement at the onset of the paragraph stating that Bird (2003) was one of two approaches used to fit the stress field (not the plate velocity field).

P3, L 5 – The paragraph that starts on this line gives many details of the calculations and what was found by them, but these details don't seem necessary for the introduction, and they obscure the description of the main point of what the paper is trying to accomplish.

P6, L 5 – Why does the model have to be run for 0.5 Myr? It seems to me that the authors are basically doing an instantaneous flow calculation, and so advancing in time is not necessary. If the 5 kyr timesteps serve as a way to iterate on the consistency between the upper and lower models, then the authors should state this (and there would be no need to advance the model for any particular length of time).

P8,L24 – the authors call a correlation of 0.82 as "relatively low" but it doesn't seem too much different than the correlation of 0.85 which was viewed more favorably earlier.

Fig. 3a – there is no scalebar for this figure to show how colors relate to geoid height.

P10, L15 – many of the setup details in this section seem to be repeated from previous sections.

Figure 8a – I think this figure is unnecessary, as the data of the WSM have been published elsewhere and are repeated in the interpolation shown in Fig. 8b. Getting rid of Fig. 8a would allow Figures 8b and 8c to be larger, which make the figures easier to see and compare.

P 17 – Some of the discussion on this page about the comparison of the modeled and

predicted stresses is more easily discussed in section 3.5.1, which refers to Fig. 9. I suggest to condense the second paragraph on page 9 and combine it into section 3.5.1, which would shorten this section on the overall comparison.

Figures 10 and 11 – Most of the information in these figures is already in Fig. 9, so perhaps these regional details are not necessary? I also think that the discussion of these figures could be shortened.

Fig. 13 – the background grid in this figure is 0.5 harmonic degrees, which is unnecessary. It would make more sense to use 1.0 degrees for the background grid (since half degrees are unphysical).

---

## Editor Comment (EC1) · T. Gerya (Editor) · 15 Jan 2018

This is an interesting and timely paper presenting results of modeling global topography and stress distribution based on a coupled numerical model of mantle convection and lithospheric dynamics. The upper 300 km shell of the model with free surface is modeled using realistic visco-elasto-plastic rheological model for the mantle and crust. The paper is of broad interest but I think the quality of presentation could be improved by addressing several discussion points listed below. Taras Gerya, Zurich, 15.01.2018

Specific points Page 1. We show that lateral density heterogeneities in the upper 300 km have a limited influence on the modeled horizontal stress field as opposed to the resulting dynamic topography that appears more sensitive to such heterogeneities. There

is hardly any difference between the stress orientation patterns predicted with and without consideration of the heterogeneities...". This low sensitivity in term of stresses seems unfortunate. Is the re any way to increase sensitivity? Changes in the dynamic topography should typically result in notable changes of bending stresses inside plates. Perhaps the method of comparing simulated and observed stresses should somehow try to isolate better the bending stress component?

Page 1. "After correction for the chemical depletion of continents, the TM2 model leads to a much better fit with the observed residual topography giving a correlation of 0.51 in continents, but this correction leads to no significant improvement in the resulting lithosphere stresses." Same as above. Would be good to understand better where major discrepancies for stresses are coming from – missing slabs? data inaccuracy?

Page 3. 2016). "The residual topography is here defined as the observed topography corrected for the variations in the crustal and lithosphere thickness and density variations and for subsidence of the sea floor with age." One could also mention here strong influence of the complex brittle–ductile rheology and stratification of the continental lithosphere result in short-wavelength modulation and localization of deformation induced by mantle flow (Burov & Guillou-Frottier, 2005).

Page 6. "A forward model is run for half a million years with a time step of 5kyr, 5 and at each time step tractions in the lower mantle due to density heterogeneities are computed using the spectral mantle code and then passed across the coupling dynamic boundary to the top component SLIM3D. Within the upper domain (SLIM3D), the flow velocities are then computed and passed back across the coupling boundary as an upper boundary condition to the spectral mantle code, with the method convergence estimated by comparing the velocity and traction norms of two successive iterations." This approach does not seem to account for continued slabs crossing 300 km depth level. It has been demonstrated by Stadler et al., (2010) that having such continued slabs is essential for properly reproducing surface plate motions.

Page 6. "Within the upper mantle, our crustal rheology is taken from Wilks (1990) and below the crust we have considered dry and wet olivine parameters in the lithosphere and sub-lithospheric mantle layers, respectively, modified after the axial compression experiments of Hirth and Kohlstedt (2004) (shown in the appendix, Table A1. Adopted from Osei Tutu et al. (2017) for studying the influence of both the driving and resisting forces that generate global plate velocities and lithospheric plate net rotation)." Reference to Osei Tutu et al. (2017) is for a paper in revision, which is not accessible.

Page 6. "Here the topographic signal induced by the layers below 300 km is assumed to be due to convection in the viscous mantle, although cold rigid subducting slabs (Zhong and Davies, 1999; Faccenna et al., 2007) and possibly also the deepest cratonic roots (Conrad and Lithgow-Bertelloni, 2006) extend deeper than 300 km." Do you account for slabs at <300 km depths? Does not seem to be the case in Fig.2, but Fig. 3d shows some slab-like features along the western margin of South America. What moves plates in the absence of slabs and prescribed surface velocities (free surface boundary condition is used) – mantle drag only? How realistic is this approach for the global plate tectonics of modern Earth, which is assumed to be predominantly driven by the slab pull? Would be good to discuss this in some more details.

Page 10. "Since the focus of this study is to investigate the effect of the upper mantle lateral density variations on the horizontal stress field and dynamic topography, an assessment of the influence of the plate boundary friction and water content in the asthenosphere on plate velocities has been carried out in a separate study (Osei Tutu et al., 2017). Hence, in the present work, we constrain our resulting creep viscosity with a cutoff for extreme viscosity values in the upper mantle by setting permissible minimum and maximum viscosity values similar to Becker (2006) and (Osei Tutu et al., 2017), with this approach yielding a good fit between the observed and modeled geoid." Would be good to give some summary of plate velocity modeling results since the referred paper (Osei Tutu et al., 2017) in review is not accessible. For example, Stadler et al. (2010) suggested that prescribing slabs in the upper mantle is essential

to reproduce global plate motions. Can you confirm this?

Page 15. "Cammarano et al. (2011) showed that correction for the depletion of the lithosphere increases the inferred temperature of a cratonic root by about 100K and decreases density by about 0.1 gcm$-3$, and fits observations well compared to models assuming pyrolitic composition." Depletion-related density decrease of the cratonic mantle is age-dependent and increase from 30 to 80 kg/m$^3$ (i.e., 0.03-0.08 g/cm$^3$) with increasing the age from the Phanerozoic to the Archean (Djomani et al., 2001)

Page 17. "We predict normal faulting mostly in regions above upwellings (mostly extensional regions) such as the Icelandic swell, Eastern African rift, or along divergent plate boundaries, while thrust faults are mainly predicted in compressional regions such as subduction zones and some other tectonically active regions in continents. In continental areas, few regional variations occur in South America, West Africa and on the Eurasian cratons. In oceans we see variations in the North Atlantic around the Icelandic swell, at the east Pacific Rise and around the southern African plate region." Strong compression seems to be predicted in the extensional backarc of the IBM subduction system (Fig. 6) – this seems problematic to me. Perhaps having continued deep and dense slabs in this region would change this?

References

Burov, E. & Guillou-Frottier, L. The plume head-lithosphere interaction using a tectonically realistic formulation for the lithosphere. Geophys. J. Int. 161, 469–490 (2005). Djomani, Y.H., O'Reilly, S.Y., Griffin, W.L., Morgan, P., 2001. The density structure of subcontinental lithosphere through time. Earth and Planet Science Letters 184, 605–621. Stadler, G., Gurnis, M., Burstedde, C., Wilcox, L.C., Alisic, L., Ghattas, O., 2010. The dynamics of plate tectonics and mantle flow: from local to global scales. Science 329, 1033–1038.

---

## Author Comment (AC1) · 15 Feb 2018

**Author's reply to:**
**"Effects of upper mantle heterogeneities on lithospheric stress field and dynamic topography" by Anthony Osei Tutu et al.**

**Referee #1 - Anonymous**

This paper presents a new mantle flow model that couples global mantle flow (driven by density heterogeneity inferred from seismic tomography) to a detailed surface model for the upper 300 km of the mantle – this includes more detailed heterogeneity associated with upper mantle viscosity variations and density heterogeneity. The authors use this coupled model to predict both lithospheric stresses and dynamic topography, which are both the result of convective stresses from viscous flow in the mantle. The authors use 2 different predictive flow models (TM1 based on heat flow and seafloor ages and TM2 based upper mantle tomography model) and compare them to the global stress map for stresses and 2 different predictions of dynamic topography. They find that both predictive flow models (which differ only in the upper 300 km) predict similar stress fields but TM2 gives a better prediction of dynamic topography. The authors conclude that dynamic topography is more sensitive to upper mantle structure than lithospheric stresses, that their model does a reasonably good job of predicting the lithospheric stress field, and that dynamic topography is better-predicted using seismic tomography data instead of heat flow and seafloor ages.

I think that the content of the paper useful and innovative – the new models are innovative and are – to my knowledge – the first of their kind to look at the impact of detailed upper mantle heterogeneity on stresses and dynamic topography. The predictions they make are useful for understanding the problem, and should lead to greater understanding of the system a. Thus, I think that the paper is eventually publishable.

We are very thankful to Referee #1 for his/her time and good recommendations and suggestions. We have attempted to address all of them in the revised manuscript as shown here below.

Ref 1: The paper has a lot of information and is a bit hard to follow because the authors are comparing 2 predictive models (TM1 and TM2) with 2 output fields (stresses and dynamic topography), one of which has 2 different options (dynamic topography from Steinberger 2016 or Hoggard 2016). Thus, there are several combinations of comparisons, and it is a bit of an effort to follow what is being compared with what. On top of that, the writing is a bit wordy with some repetition. This makes reading and understanding the paper a bit laborious.

I think for this paper to be effective, it needs better organization and structure, and the writing needs to be simplified and shortened. I give some suggestions for this below. Given this, I recommend significant re-revision of the figures and text, although I do not think that significant new analysis is necessary. Thus, I would probably call for "moderate revision". Below is a an overall assessment of the changes that I think would be useful for improving the paper and also some specific points about the paper.

AU: This is a very important suggestion, on the readability and understanding of our manuscript. Hence, we follow the referee's suggestions and have also re-organized the manuscript where it's necessary for easier reading.

**Major Points:**

Ref 2:
**Overall Structure**: One problem is that the predictions and analysis of stresses and dynamic topography are intermingled in section 3. I recommend separating this into two sections, one that is devoted to stresses and the other to dynamic topography. Thus, I recommend removing the topography predictions from Figs. 5 and 6 (so these figures are devoted to stresses), and removing the stresses from Fig. 7 (so this figure is for dynamic topography). I also recommend revising the text around these figures to focus specifically on stresses and then on dynamic topography.

AU: We have re-organized the manuscript to separate the reconstruction of the lithosphere stress field and dynamic topography together with the discussions around these reconstructions. The corresponding figures are also re-organized. Where necessary for the benefit of comparison and discussions with regards to the focus of the study, we have included and explained figures in the authors' response.

Ref 3:
**Introduction** – I think the introduction is a bit too long and rambling, and should get to the main point more quickly – that this paper presents a new method for evaluating the role of upper mantle heterogeneity (both density heterogeneity and viscosity heterogeneity) for mantle stresses. There is much background – condensing it to just the relevant parts will help the authors emphasize their new contribution a bit more clearly.

AU: We agree with this referee's suggestion to shorten the introduction and emphasize on the new approach our study presents. We have revised the introduction to make it short and concise with the focus on what is new with our study.

"See the attached tracked manuscript below"

Ref 4:
**Conclusions** – I think that the paper does a nice job of concluding the major results. But there is no discussion section – it would be helpful to have a short discussion of the implications of this work for our understanding of dynamic topography and stresses, and other factors that might be important to include in the models in the future.
AU: Since most of the discussions on the stress field and dynamic topography are presented in the respective extended sections in the chapter "Results and Discussion", we thought to rather give an extensive conclusion on the results.

**Figures**-There are more figures than are necessary. In particular:

Ref 5:
Figure 1 and 3 are illustrative about the model, but the information in these figures is not used much – perhaps only the essential components of Fig. 3 could be incorporated into Fig.1? (for example, the geoid, plate velocity, and tractions are not really used here)

AU: We combined the most relevant Figures in Figures 1 and 2 and also in Figures 3 and 4 resulting in the current new Figures 1 and 2 respectively, as shown below:

[Figure]

**Figure 1**: Adopted from Osei Tutu et al., (2017) (a) Depth-dependent scaling profile of S-wave velocity to density; (b) radial mantle viscosity structure (Steinberger and Calderwood, 2006) and (c) a schematic diagram of the numerical method that couples the 3D-lithosphere-asthenosphere code SLIM3D (Popov and Sobolev, 2008) to a lower mantle spectral flow code (Hager and O'Connell, 1981) at a depth of 300 km. The 3D thermal structure of the upper mantle (i.e. 300 km) at a depth of 80 km from two reference thermal models adopted in this study. d) TM1, a heat flow-based thermal structure inferred from the TC1 model of Artemieva (2006) in the continents and the sea floor age model of Müller et al. (2008) in the oceanic areas. e) TM2, the thermal structure of the upper mantle inferred from the S-wave tomography model SL2013sv of Schaeffer and Lebedev (2013). The "ring-ing" visible in the upper structure is a side effect introduced by smoothing sharp boundaries with a spherical harmonic expansion.

[Figure]

Figure 2: Modeled geoid anomaly and (b) modeled plate velocity, considering lateral viscosity variations with the TM1 thermal-density model in the upper 300 and a 3-D density structure of the mantle inferred from Becker and Boschi (2001) in combination with the layered viscosity profile from Steinberger and Calderwood (2006) imposed below 300 km. (c) Resulting total shear tractions at the 300 km depth generating stresses in the crust and lithospheric depth. (d) The corresponding average creep viscosity versus depth in the upper mantle (300 km) across continents and oceans, considering different olivine parameters.

NB:
Even though, Figure 2 is just illustrative of our model fitting other geophysical observables such as the geoid and plate motions (Figure 2a-b). Here also we maintain the figure with tractions to refer from the main text to what causes/generates the stresses in the lithosphere due to the viscous mantle flow.

Ref 6:

Figure 4 shows comparisons for different models of dislocation and diffusion creep, and different water contents – however it is not clear to me that all of this complexity is necessary? Why not simplify by showing only the most relevant models and say in the text what is the impact of changing the rheological parameters?

AU: We have combined the average resulting depth dependent creep viscosity into the geoid, plate motions and tractions figure. Shown above in Figure 2.

Ref 7:
As I mentioned above, I think that figures 5cd, 6cd, and 7cd are not necessary. Fig. 7 should be moved later into the dynamic topography section.

AU: We agree with the suggestion related to the previous version of Figures 5cd and 6cd, which have been moved to the supplementary information and are referred to

from the manuscript. However, we consider that Figures 7c-d (now Figures 8c-d) are important for carrying out our analysis of the influence of the crust on the stress field as compared to Figures 3&4(a-b) when we discuss the modeled topography.

[Figure]

Figure3: a) Model-based maximum horizontal stress magnitude and most compressive stress directions [SHmax] following the convention with compression being positive, originating from mantle flow driven by density anomalies below 300 km. (b) Same for structure of the top 300 km of the upper mantle, computed with the CRUST 1.0 model and TM1 beneath air (free surface)

[Figure]

Figure 4: Predictions of the SHmax magnitude and direction from combined contributions due to lower mantle flow and upper mantle from a) TM1 with crust model and b) TM2 with crust model.

Ref 8:
For the stress comparisons, I mention below that Fig. 8a is not necessary. Also Figures 10 and 11 could probably be combined.

AU: The original Figure 8a has been removed, since it's readily available from the World Stress Map repository; so this allowed us to increase the size of the remaining two figures. The regional comparisons have also been combined into one figure, and this is included and further discussed to show how the choice of the upper mantle structure (how well the data source is constrained) influences the model fit to the

WSM2016. AU: Hence, we have excluded Figure 8a and enlarged Figures 8b and 8c (now Figures 5a and 5b) as shown below:

[Figure]

Figure 5: Interpolated World stress map 2016 (Heidbach et al., 2016), data on a grid of 2.5° x 2.5°, using only stress orientation with a constant search radius 270 km, and (b) predicted SHmax orientation and regime from total stress contribution with TM1 (plotted in thin bars) over TM2 (thick bars) upper mantle thermal structures. Colors of dots (a) and bars (b) indicate observed or predicted stress regime with red for normal faults or tensile stress, blue for thrust faults or compressive stress, and green for strike-slip faults or intermediate stress (one principal horizontal stress positive, one negative)

We have combined Figures 10 and 11 as shown below at response to Ref 23.

Ref 9:
Figures 7ab and 12cd both show dynamic topography for models TM1 and TM2, the only difference being that the effect of seafloor age is removed from 12cd. I think that it is appropriate to only use Figs. 12cd and omit 7ab, because the models 12cd are compared to the "observation" models 12ab.

AU: We would like to keep the modeled dynamic topography and the resulting stress field shown in Figure 7 (now Figure 8) to support our discussion on the wide amplitude variations in the modeled dynamic topography due to the isostatic effect of the sea floor cooling, which does not translate into significant variations in stress pattern and also in the scenarios where crustal structure is not considered in continents.

Ref 10:
Figure 13 – I am a little unsure how the amplitudes of spherical harmonic degrees are compared for only parts of the globe (continents or oceans). The spherical harmonics are globally-defined, so does it make sense to compare the spherical harmonic components over the oceans-only or continents-only? How is the rest of the world considered when the spherical harmonic components of the rest of the world are defined? I wonder if it might be better to just do the comparison for the whole globe and avoid this problem?

AU: We have included a discussion of this issue in the manuscript. We assigned the global mean value in continents when computing the ratio for oceans and vice versa. When the same exercise is done with zeros in continents to estimate the oceans ratio and vice versa, there is no change in the shape of the curves but rather, a shift in the vertical axis. To understand the significance of regional features, the contribution of each region at different wavelengths is of particular importance here, but the approach of global ratio or amplitude spectrum for continents and oceans together is not easily distinguishable. With this approach we show that the dominant contribution in the ocean spectrum is between spherical harmonics degrees 1 to 2 with less variability at higher spherical harmonics. However, in continents, we see a wider variability at higher spherical harmonic higher (up to degree 15), mainly coming from cratons, which will guide the corrections for chemical depletion in continental regions.

Ref 11:
Figure 14 – perhaps could be added as an alternative predictive model to Fig. 12?

AU: Done

[Figure]

Figure 9: Comparing a) the in-situ observed residual topography from Hoggard et al., (2016), and b) the residual topography based on the CRUST 1.0 from Steinberger, (2016) with modeled dynamic topography using TM1 (c) and TM2 (d) upper mantle thermal density structures with the effect of the sea floor cooling with age removed. Green dots with black circles around show locations of major hotspots

**Specific points**:

Ref 12:
Title – I think the word "the" should be added before "lithospheric stress field"
AU: The revised title now reads

"Effects of upper mantle heterogeneities on the lithospheric stress field and dynamic topography"

Ref 13: Page 1 – the statements on lines 7-8 and 9-11 basically say the same thing – I think this sort of repetition is not needed in the abstract

AU: The statement spanning lines 9-11 has been removed and an additional revised statement below included:

Ref 14: Page 1, line 12 – stating that a correlation has a value of 0.51 doesn't mean too much to a reader – is this correlation good or poor?
AU: The revised statement reads:

…..giving a good correlation of 0.51 in continents….

Ref 15: P1, L13 – it is not the lithospheric stresses that are being improved, it is the model fit to them.
AU: We have reformulated the statement as:

"……fit between the WSM2016 and the resulting lithosphere stresses."

Ref 16: P1, L15 – the difference in angular misfits reported here (18.3 vs 19.9 deg) doesn't seem very significant – I'm not sure that it is useful to include this information Abstract – generally, it would be good to close the abstract with an overall general statement about what the reader should take away from this study. Currently the abstract doesn't do this.
AU: We revised the abstract to include the statements below:

"Our findings show the disparity of the contributions coming from the shallow and deep mantle dynamic forces to lithosphere stress field and dynamic topography."….

Ref 17: P2, L 27 – this paragraph starts "At longer wavelengths," which seems to differentiate it from the previous paragraph. But both paragraphs are about long wavelengths. It would be better to distinguish this paragraph by stating out the onset that it deals with dynamic topography.

AU: We agree that there is a need to highlight the contrast between the two paragraphs at Lines 15 and 27 regarding the lithosphere stress field and topography. Nonetheless, we briefly discussed the crustal contribution at short wavelengths to the stress field at the end of the former paragraph (Lines 15-26). The revised opening of the later paragraph (Line 27) reads:

"Likewise, the long wavelength signal of topography is related to the vertical component of the stress field tensor originating from the thermal convection of the mantle rocks (Pekeris, 1935; Steinberger et al., 2001)"

Ref 18: P3, L 27 – Here the Bird (2003) approach is described as using the fit to the observed plate motions – this needs a clearer explanation because it contradicts the statement at the onset of the paragraph stating that Bird (2003) was one of two approaches used to fit the stress field (not the plate velocity field).

AU: There seems to be confusion here with regards to our reference to Bird (2003) and Bird et al. (2008). We used the digital model of plate boundaries to predefine our modeling plate boundaries of Bird (2003) to predefine our modeling plate boundaries in the setup as mentioned in the appendix.

With regards to Bird et al. (2008), their study estimates lithosphere stress field, strain rate, plate velocity and seismic anisotropy, using constraints from boundary conditions, variable crust and lithosphere thicknesses, nonlinear rheology and the fit between the modeled and observed plate velocities, by disregarding the effect of deep mantle flow. This approach seemingly works well compared to other studies that have considered the contribution from the deep mantle in addition (Ghosh and Holt, 2012;

Steinberger et al., 2001; Lithgow-Bertelloni and Guynn, 2004). We revised the sentence as below:

"On the other hand, Bird et al. (2008) have estimated the lithospheric stress from a model that disregards the mantle flow contribution and used the fit between the modeled and observed plate velocities as a sole criterion"

Ref 19: P4, L 5 – The paragraph that starts on this line gives many details of the calculations and what was found by them, but these details don't seem necessary for the introduction, and they obscure the description of the main point of what the paper is trying to accomplish.

AU: We agree with this suggestion, since most of these statements are already mentioned in the method and result sections that follow. We have reformulated this part of the introduction as shown below:

"To date, two distinct approaches have been adopted to study the origin of the lithospheric stress, and each has given a relatively good fit to the observed stress field. On the one hand, Bird et al. (2008) have estimated the lithospheric stress from a model that disregards the mantle flow contribution and used the fit between the modeled and the observed plate velocities as a sole criterion}. On the other hand, Ghosh et al., (2013), Ghosh ad Holt (2012), Lithgow-Bertelloni and Guynn (2004), Steinberger et al., (2001) and Wang et al., (2015) have aimed at assessing the influence of the mantle flow on the lithospheric stress field and have shown that the bulk mantle flow explains a large part (about 80-90 %) of the stress field accumulated in the lithosphere Steinberger et al., (2001), in both magnitude and the most compressive horizontal direction. The aim of the present study is to evaluate the contribution of the upper mantle density and viscosity heterogeneities above the transition zone to the observed spatial stress regimes of the lithosphere (Heidbach et al., 2016), while testing different approaches and data sets used to describe the thermal and rheological structure of the upper mantle and the crust. We use a 3D global lithosphere-asthenosphere finite element model (Popov and Sobolev 2008; Sobolev et al., 2009) with visco-elasto-plastic rheology, which is coupled to a spectral model of mantle flow (Hager and O'Connell, 1981) at 300 km depth. Deriving all force contributions from a single calculation resolves any inconsistency that might arise from treating individual force contributions to the stress field separately, as has been done in earlier studies (Bird et al, 2008; Steinberger et al 2001; Lithgow-Bertelloni and Guynn, 2004; Ghosh et al., 2008; Naliboff et al., 2012; Ghosh et al., 2013; Wang et al., 2015). As part of this work, we further estimate dynamic topography and correlate our results with two different residual topography models. One is based on seismic surveys of the ocean floor used to correct for shallow contributions to topography and free-air gravity anomalies on continents Hoggard et al., (2016). The second model is taken from Steinberger (2016) and is based on actual topography corrected for crustal thickness and density from CRUST1.0 (Laske et al., 2013). "

Ref 20: P6, L 5 – Why does the model have to be run for 0.5 Myr? It seems to me that the authors are basically doing an instantaneous flow calculation, and so advancing in time is not necessary. If the 5 kyr time-steps serve as a way to iterate on the consistency between the upper and lower models, then the authors should state this (and there would be no need to advance the model for any particular length of time).

AU: The study results are compared with present-day observables (WSM2016 and residual topography), which is why we have restricted our calculations to half a million of years.

\*Correction->There is also a typographical error in our time-step; it's 50 kyr not 5kyr.

Yes, you are right that using 50-kyr time-steps allows us to track our iterations between the two-coupled components for consistency. As we have mentioned in the manuscript (Page 6, Line 30) this is done by comparing the norms of the velocities and tractions from two successive iterations.

Ref 21: P8, L24 – the authors call a correlation of 0.82 as "relatively low" but it doesn't seem too much different than the correlation of 0.85, which was viewed more favorably earlier. Fig. 3a – there is no scale bar for this figure to show how colors relate to geoid height.

AU: We report this correlation difference for the modeled geoid with and without lateral viscosity variations (LVVs) in the top 300 km to highlight, how the global correlation up to spherical harmonic degree 31 alone does not show the improvement in the fit to the observed geoid. However, looking at the degree-by-degree correlation and amplitude spectrum gives us more information about contribution of lateral viscosity in the top 300 km, as shown in the Figure 2R below. This support the conclusion of by Čadek and Fleitout (2003) about the importance of LVVs in the top 300 km. The figure below is included the supplementary of Osei Tutu et al. (2018). Also we have corrected the geoid figure by including the colorbar shown above in Figure 2 response to Ref 4.

[Figure]

Figure 2R: Estimates of correlation and amplitude spectrum of modeled and observed geoid for spherical harmonic degrees l =1to 30

Ref 23: P10, L15 – many of the setup details in this section seem to be repeated from previous sections.

AU: We revised the beginning of the paragraph as below to avoid the repetition of the model set up:

"We start with examining separate contributions of the mantle heterogeneities below 300 km (deep Earth setup) and above (shallow Earth setup) to the global lithospheric stress field and topography. To calculate the contribution of the lower domain, we use a constant lithosphere thickness (100 km) and density (3.27 kg/m³), with a configuration of the mantle below similar to that used to derive the geoid anomaly, plate motions and the shear tractions in Figure 3a-c"

Figure 8a – I think this figure is unnecessary, as the data of the WSM have been published elsewhere and are repeated in the interpolation shown in Fig. 8b. Getting rid of Fig. 8a would allow Figures 8b and 8c to be larger, which make the figures easier to see and compare.

AU: Done in above at figure at response to Ref 8.

Ref 22: P 17 – Some of the discussion on this page about the comparison of the modeled and predicted stresses is more easily discussed in section 3.5.1, which refers to Fig. 9. I suggest condensing the second paragraph on page 9 and combining it into section 3.5.1, which would shorten this section on the overall comparison.

AU: We have combined the second paragraph of section 3.5 and combined with section 3.5.1 and the regional comparison, which is highlighted in red in the tracking manuscript and shown here below:

"See the attached tracked manuscript below"

Ref 23: Figures 10 and 11 – Most of the information in these figures is already in Fig. 9, so perhaps these regional details are not necessary? I also think that the discussion of these figures could be shortened.

AU: These regional detail comparisons are considered in order to discuss the implications of the continental thermal structures used, their strengths and weaknesses and influence on the model stress field on regional basis. The figures of the angular misfit for Western Europe and Australia are discussed in detail to show the impact of the well-constrained thermal structures in these regions on the lithosphere stress field and the fit to the observed stress field. We as well point out how treating slabs also influence the fit in subduction regions, as given in the revised manuscript and the accompanying tracking manuscript.

[Figure]

Figure 7: Regional comparison of the angular misfit in Europe (a and b) and Australia (c and d) between the observed and modeled total stresses with TM1 (a-c) and TM2 (d-f). Red bars denote modeled orientations versus black bars showing the smoothed observed stress field (WSM2016).

Ref 24: Fig. 13 – the background grid in this figure is 0.5 harmonic degrees, which is unnecessary. It would make more sense to use 1.0 degree for the background grid (since half degrees are unphysical).

AU: We have re-plotted figure 13, now as figure 11, using background grid interval of spherical harmonic degree 1 as shown below.

[Figure]

Figure 10: Ratio of modeled dynamic topography from TM1 and TM2 for (a) continents and (b) oceanic regions with observations based residual topography from Steinberger (2016) and (Hoggard et al., (2016)

*density value of 0.04 gcm$^{-3}$ (about 300 K) as correction in TM2 cratons is considered to compare with models with realistic corrections. 
[revised manuscript text omitted]

---

## Author Comment (AC2) · 15 Feb 2018

**"Effects of upper mantle heterogeneities on lithospheric stress field and dynamic topography"**
**by Anthony Osei Tutu et al.**

**Referee #2 T. Gerya (Editor) taras.gerya@erdw.ethz.ch**

This is an interesting and timely paper presenting results of modeling global topography and stress distribution based on a coupled numerical model of mantle convection and lithospheric dynamics. The upper 300 km shell of the model with free surface is modeled using realistic visco-elasto-plastic rheological model for the mantle and crust. The paper is of broad interest but I think the quality of presentation could be improved by addressing several discussion points listed below.
Taras Gerya, Zurich, 15.01.2018

We are very much appreciate Taras Gerya time taken to look at our manuscript, the comments and suggestions to help us improve the manuscript. In the paragraphs below, we have carefully considered each comment and provided the response. Also we have accounted for the required modification in the revised manuscript were relevant.

Ref. Points 1-Page 1:
We show that lateral density heterogeneities in the upper 300 km have a limited influence on the modeled horizontal stress field as opposed to the resulting dynamic topography that appears more sensitive to such heterogeneities. There is hardly any difference between the stress orientation patterns predicted with and with- out consideration of the heterogeneities. . .". This low sensitivity in term of stresses seems unfortunate. Is there any way to increase sensitivity? Changes in the dynamic topography should typically result in notable changes of bending stresses inside plates. Perhaps the method of comparing simulated and observed stresses should somehow try to isolate better the bending stress component?

Although the overall lower sensitivity of the modeled lithospheric stress field to the upper mantle heterogeneities (here the top 300 km) marks one of the findings of this study, we find some significant regional influences such as in the Andes, due to the crustal thickness variations and upper mantle heterogeneities. These regional effects together with global stress pattern when compared to the observed stress show the range of impacts of the deep mantle flow.
We also find that using different data to describe the upper mantle structure, either based on seismic topography or heat flow data, did not significantly impact stress field magnitudes and horizontal directions. This is illustrated in Fig. R1 where we estimate the differences in the respective modeled stress magnitudes and orientations as well as their corresponding dynamic topographies. In Fig. R1 (a & b) we show the differences between the modeled stress magnitude with crustal thickness variations (Fig R1a) and without (Fig. R1b) and the corresponding stress orientations differences (fig R1c and d).
However, in Fig. R1e the difference in dynamic topography models shows very large amplitudes in cratons, which is not the case for the corresponding differences in the stress magnitudes (fig R1b) or orientations (fig. R1d). This may suggest that changes

in topography may not readily translate into bending stresses inside the plate interior. Also, as we mentioned in the study lateral viscosity variations in the crust and lithosphere could be one major influence on the stress sensitivity controlling how the lithosphere strength responds to the mantle flow below. However, in this study we concentrate on the thermal-density structure of the upper mantle without exploring variations in the strength of the lithosphere and a rheological parameter space to understand how the stress will be transmitted elastically over long distances.

A study of different crustal and lithosphere lateral viscosity variations should be the focus of a future study, now that the dependency dependence of the stress field on the upper mantle density heterogeneities is established.

[Figure]

Figure 1R: Estimates of the differences between modeled stress (a) magnitudes and (c) orientation of main manuscript Figure 6 (a&b). Also similar exercise is shown in (c) magnitude difference and (d) orientation difference of the main manuscript Figure 7 (c & d). In (e) is the corresponding dynamic topography difference between main manuscript Figure 7a and 7b.

Ref. Points 2-Page 1:
"After correction for the chemical depletion of continents, the TM2 model leads to a much better fit with the observed residual topography giving a correlation of 0.51 in continents, but this correction leads to no significant improvement in the resulting lithosphere stresses." Same as above. Would be good to understand better where major discrepancies for stresses are coming from – missing slabs? data inaccuracy?

To add to the above response, Naliboff et al., (2009) showed the influence of the cratonic roots on lithosphere stress field in both magnitude and pattern is small compared to cratonic influence on the changes in mantle tractions generating the stresses in the lithosphere plate. The regional influence on topography due to realistic treatment of the cold craton is not apparent in the respective stress field probably due to the integrated effect of large mantle driving forces transmitted elastically from far field through the lithosphere and thus overwhelming the local change in the mantle lithospheric structure.

Taking for instance, the IBM region, when we consider lithosphere stresses due to only the viscous mantle flow, we a predict purely compressional regime (Figure 3a). However, in the scenario considering only the crust, the lithosphere and a part of the asthenosphere above 300 km (with muted contribution of the lower mantle), we predict an extensional stress regime for the IBM subduction region (Figure 3b). Their combined contributions to the stress field make the IBM regions compressional, showing the influence of the integrated traction from the viscous mantle. When we explicitly include slabs in the top 300 km (TM1), the compressional regimes overwhelms the observed extensional back-arc IBM subduction system but the stress pattern does not seem to significantly change. * We further comment on this in Ref. Points 9-Page 17*

We can attribute some of the discrepancies between the model and the observational data (WSM2016) to the interpolation method, the search radius for interpolation and the upper mantle structure considered. However, there is a relatively good agreement in most regions between the modeled and observed stress fields with some misfit in some regions due to the thermal density structure considered in the upper mantle. For instance, in the Tibet region, considering heat flow-based thermal structure (TM1) give a better fit for the to the observed stress field compared to the modeled stress field with the S-wave model. Also, the inclusion of slabs in TM1 gives a better fit in the Sumatra subduction compared to the slabs implicitly included in the seismic model (TM2).

There are some regions such as the Colorado Plateau that will still need further investigation with regards to the disorientation of the stress patterns between the model and observations. This will be appropriate for future studies using recent high-resolution seismic tomography from the US Array to help understand that anomaly.

Ref. Points 3-Page 3:
"The residual topography is here defined as the observed topography corrected for the variations in the crustal and lithosphere thickness and density variations and for subsidence of the sea floor with age." One could also mention here strong influence of the complex brittle–ductile rheology and stratification of the continental lithosphere result in short-wavelength modulation and localization of deformation induced by mantle flow (Burov & Guillou-Frottier, 2005).

Yes, this is very true with regards to topographic change in the lower crust and the lithosphere. We have included the sentence below:

"Also at the tectonic-scale topography is influenced by the elastic-brittle–ductile layered crustal-lithospheric layered structures underline by the viscous mantle convection (Burov and Guillou-Frottier 2005)."

NB: The introduction paragraph containing "The residual topography is here defined as the observed topography corrected for the variations in the crustal and lithosphere thickness and density variations and for subsidence of the sea floor with age." has been removed per the suggestion of the first referee to shorten the introduction.

Ref. Points 4-Page 6:
"A forward model is run for half a million years with a time step of 5kyr, 5 and at each time step tractions in the lower mantle due to density heterogeneities are computed using the spectral mantle code and then passed across the coupling dynamic boundary to the top component SLIM3D. Within the upper domain (SLIM3D), the flow velocities are then computed and passed back across the coupling boundary as an upper boundary condition to the spectral mantle code, with the method convergence estimated by comparing the velocity and traction norms of two successive iterations." This approach does not seem to account for continued slabs crossing 300 km depth level. It has been demonstrated by Stadler et al., (2010) that having such continued slabs is essential for properly reproducing surface plate motions.

AU: Here, in our calculation within the mantle above the 300 km we explicitly consider slabs in the TM1 thermal structure and below 300 km depth we assume density heterogeneities corresponding to slabs and upwellings captured by the Smean seismic tomography without explicitly including slabs. Furthermore, in our plate motion paper (Osei Tutu et al. 2018), we show that with this approach when used to predict global plate motions, we obtained a good fit to the observed plate motions NUVEL-1A (DeMets et al., 2010) with rms velocity of 3.75 cm/yr.

Ref. Points 5-Page 6:
"Within the upper mantle, our crustal rheology is taken from Wilks (1990) and below the crust we have considered dry and wet olivine parameters in the lithosphere and sub-lithospheric mantle layers, respectively, modified after the axial compression experiments of Hirth and Kohlstedt (2004) (shown in the appendix, Table A1. Adopted from Osei Tutu et al. (2017) for studying the influence of both the driving and resisting forces that generate global plate velocities and lithospheric plate net rotation)." Reference to Osei Tutu et al. (2017) is for a paper in revision, which is not accessible.
AU: This referenced referred paper (Osei Tutu et al. 2018) is now published. We presumed it would be out before the review process of this paper has been finished; that is why we referred to it instead of repeating the exercise in this manuscript.

Ref. Points 6-Page 6:
"Here the topographic signal induced by the layers below 300 km is assumed to be due to convection in the viscous mantle, although cold rigid subducting slabs (Zhong and Davies, 1999; Faccenna et al., 2007) and possibly also the deepest cratonic roots (Conrad and Lithgow-Bertelloni, 2006) extend deeper than 300 km." Do you account for slabs at <300 km depths? Does not seem to be the case in Fig.2, but Fig. 3d shows some slab-like features along the western margin of South America. What moves plates in the absence of slabs and prescribed surface velocities (free surface boundary condition is used) – mantle drag only? How realistic is this approach for the global

plate tectonics of modern Earth, which is assumed to be predominantly driven by the slab pull? Would be good to discuss this in some more details.

AU: Yes, we account for slabs at depths above 300 km in the TM1 upper mantle structure while we assumed that slabs are picked by the seismic tomography the S-wave structure TM2. Likewise in previous Figure 3d (now Figure 2c) the slab signal under South America is coming from the Smean tomography without explicitly including slabs below the 300 km. As we have shown for different depth slices for TM1 on the left column.

[Figure]

Figure S1: The thermal structure of the upper mantle at a depths of 35, 100, 150, and 250 km from the two reference thermal models adopted in this study, TM1 (left column) and TM2 (right column). TM1 is derived from the thermal structure TC1 of Artemieva (2006) in the continents and the sea floor age model of Müller et al. (2008) in the oceanic areas, while the TM2 model is based on the S-wave tomography-model SL2013sv from Schaeffer and Lebedev (2013) for inferring thermal structure in the upper 300 km. A detailed description is given in the main text

Ref. Points 7-Page 10:
"Since the focus of this study is to investigate the effect of the upper mantle lateral density variations on the horizontal stress field and dynamic topography, an assessment of the influence of the plate boundary friction and water content in the asthenosphere on plate velocities has been carried out in a separate study (Osei Tutu et al., 2017). Hence, in the present work, we constrain our resulting creep viscosity with a cutoff for extreme viscosity values in the upper mantle by setting permissible minimum and maximum viscosity values similar to Becker (2006) and (Osei Tutu et al., 2017), with this approach yielding a good fit between the observed and modeled geoid." Would be good to give some summary of plate velocity modeling results since the referred paper (Osei Tutu et al., 2017) in review is not accessible. For example, Stadler et al. (2010) suggested that prescribing slabs in the upper mantle is essential to reproduce global plate motions. Can you confirm this?

AU: Our analysis in Osei Tutu et al., (2018) with explicit inclusion of slabs in the top 300 km and viscous mantle drive from seismic tomography (Smean) below confirms the suggestions of Stadler et al. (2010). In instances where slabs were not considered, the fit to the observed plate motion deteriorates. However, the benefits of considering slabs in the stress field studies are arguable. An inclusion of slabs improves the fit in the Sumatra subduction area, while contributing to a strong compression regime in the IBM extensional subduction area.

Ref. Points 8-Page 15:
"Cammarano et al. (2011) showed that correction for the depletion of the lithosphere increases the inferred temperature of a cratonic root by about 100K and decreases density by about 0.1 $gcm^{-3}$, and fits observations well compared to models assuming pyrolitic composition." Depletion-related density decrease of the cratonic mantle is age-dependent and increase from 30 to 80 kg/m^3 (i.e., 0.03-0.08 g/cm^3) with increasing the age from the Phanerozoic to the Archean (Djomani et al., 2001)

AU: Thank you for the spotting this! Indeed, there was a mistake in the density difference value, which is assumed in the study (i.e. 0.01 gcm-3) as opposed to the wrong value mentioned in the text (0.1 gcm-3), which would be very low. In our qualitative analysis we do not distinguish cratonic regions based on age progression from the Phanerozoic to the Archean, but rather considered a single value for all cratons.
We have taken your suggestion into account and use the mean value of 40 kg/m^3 (0.04 gcm-3) for a density correction of depletion-related density in cratons. We know that a proper treatment of different ages of cratons may improve the modeled topography, but our analysis shows there seems to be no significant impact on the stress field. We therefore have reformulated the relevant part as of the manuscript as:

"Previous studies of cratonic mantle depletion in- relation to density and temperature inferred from S-wave models (for example, Cammarano et al., 2011) identified composition as the key dominant agent for the low modeled topography. They showed that with the assumption of only 100 K hotter mantle combined with lateral variations in composition resulted in a difference in density of about 0.1 gcm-3 compared to models assuming pyrolitic composition. The depletion-related density in cratons is age-dependent and considered to be increasing from 30 to 80 kgm-3 (i.e. 0.03 to 0.08 gcm-3) for the Phanerozoic through Protozoic to Archean platforms ( (Djomani et al. 2001). Here we aim at a qualitative first order analysis imposing an additional mean density value of 0.04 gcm-3 (about 300 K) as a correction in TM2 cratons to compare with models using realistic corrections. Also, following the realistic compositional correction in cratons by Cammarano et al., (2011) we adopt

two additional thermal structures from different seismic tomography models SAW24B16 (Mégnin and Romanowicz 2000) and S20RTS (Ritsema et al. 2007) with corrections applied to the depleted mantle based on the thermodynamic model Perple_x (www.perplex.ethz.ch, Connolly, 2005) and compare with our results."

Our increase in density/temperature for the cratonic regions as a correction in the TM2 decreases the correlation from 0.512 with Steinberger (2016) for the 100 K to 0.50 for the 300 K considered, while the correlation with Hoggard et al., (2016) increases the correlation from 0.180 to 0.192, respectively, since the free-air gravity used by Hoggard et al., (2016) creates positive topographic anomalies across most cratons.

Ref. Points 9-Page 17:
"We predict normal faulting mostly in regions above upwellings (mostly extensional regions) such as the Icelandic swell, Eastern African rift, or along divergent plate boundaries, while thrust faults are mainly predicted in compressional regions such as subduction zones and some other tectonically active regions in continents. In continental areas, few regional variations occur in South America, West Africa and on the Eurasian cratons. In oceans we see variations in the North Atlantic around the Icelandic swell, at the east Pacific Rise and around the southern African plate region." Strong compression seems to be predicted in the extensional backarc of the IBM sub-duction system (Fig. 6) – this seems problematic to me. Perhaps having continued deep and dense slabs in this region would change this?

AU: Our inclusion of explicit slabs in the top 300 km (TM1) seems to contribute to this strong compressional regime, which is not the case when we rather consider the rather slabs captured by the s-wave model (TM2). The IBM subduction system is shown as compressional when we consider only density heterogeneities below 300 km (figure 3a), similar to what Steinberger et al., (2001) obtained for calculations with viscous mantle flow and half the speed of the free plate motion prescribed as top boundary condition. Having continued slab into depths below 300 km might influence the strong compressional regime we have predicted but our coupled numerical model accounts for realistic crustal-lithosphere structure separately from the viscous mantle and does not allow for the continuity of the slab material. Nonetheless, the implemented continuity of velocities and tractions is very robust, helping us understand stress pattern and regimes in the lithosphere as it is for plate motions.

Also, we do not account for melting and fluid releases in our calculations, which are predominantly the cause of upwelling in the IBM region due to the interaction between subducting and overriding plates. This may contribute to the extensional stress regime of the IBM subduction system (Arculus et al. 2015; Brandl et al. 2017). Hence, such study considering melting and fluid release in future probe is encouraged to shine some light on the dominance of the lower mantle compressional regime in the IBM subduction zone reported here in this study.

References:

Arculus, Richard J. et al. 2015. "A Record of Spontaneous Subduction Initiation in the Izu–Bonin–Mariana Arc." *Nature Geoscience* 8:728. Retrieved (http://dx.doi.org/10.1038/ngeo2515).

Artemieva, Irina M. 2006. "Global 1°×1° Thermal Model TC1 for the Continental Lithosphere: Implications for Lithosphere Secular Evolution." *Tectonophysics* 416(1):245–77. Retrieved (http://www.sciencedirect.com/science/article/pii/S0040195105006256).

Brandl, Philipp A. et al. 2017. "The Arc Arises: The Links between Volcanic Output, Arc Evolution and Melt Composition." *Earth and Planetary Science Letters* 461:73–84. Retrieved (http://www.sciencedirect.com/science/article/pii/S0012821X16307403).

Burov, E. and L. Guillou-Frottier. 2005. "The Plume Head–continental Lithosphere Interaction Using a Tectonically Realistic Formulation for the Lithosphere." *Geophysical Journal International* 161(2):469–90. Retrieved (http://dx.doi.org/10.1111/j.1365-246X.2005.02588.x).

Cammarano, Fabio, Paul Tackley, and Lapo Boschi. 2011. "Seismic, Petrological and Geodynamical Constraints on Thermal and Compositional Structure of the Upper Mantle: Global Thermochemical Models." *Geophys. J. Int* 187:1301–18.

Charles Mégnin  Barbara Romanowicz. 2000. "Three-dimensional Shear Velocity Structure of the Mantle from the Inversion of Body, Surface and Higher-mode Waveforms | Geophysical Journal International | Oxford Academic." *gji*. Retrieved August 29, 2017 (https://academic.oup.com/gji/article/143/3/709/719966).

DeMets, Charles, Richard G. Gordon, and Donald F. Argus. 2010. "Geologically Current Plate Motions." *Geophysical Journal International* 181(1):1–80. Retrieved January 6, 2017 (http://gji.oxfordjournals.org/cgi/doi/10.1111/j.1365-246X.2009.04491.x).

Müller, R.Dietmar, Maria Sdrolias, Carmen Gaina, and Walter R. Roest. 2008. "Age, Spreading Rates, and Spreading Asymmetry of the World's Ocean Crust." *Geochemistry, Geophysics, Geosystems* 9(4):n/a-n/a. Retrieved May 11, 2016 (http://doi.wiley.com/10.1029/2007GC001743).

Osei Tutu, A., S. V Sobolev, B. Steinberger, A. A. Popov, and I. Rogozhina. 2018. "Evaluating the Influence of Plate Boundary Friction and Mantle Viscosity on Plate Velocities." *Geochemistry, Geophysics, Geosystems* n/a-n/a. Retrieved (http://dx.doi.org/10.1002/2017GC007112).

Poudjom Djomani, Yvette H., Suzanne Y. O'Reilly, W. L. Griffin, and P. Morgan. 2001. "The Density Structure of Subcontinental Lithosphere through Time." *Earth and Planetary Science Letters* 184(3):605–21. Retrieved (http://www.sciencedirect.com/science/article/pii/S0012821X00003629).

Ritsema, J., A. Deuss, H. J. van Heijst, and J. H. Woodhouse. 2011. "S40RTS: A Degree-40 Shear-Velocity Model for the Mantle from New Rayleigh Wave Dispersion, Teleseismic Traveltime and Normal-Mode Splitting Function Measurements." *Geophysical Journal International* 184(3):1223–36. Retrieved January 4, 2017 (https://academic.oup.com/gji/article-lookup/doi/10.1111/j.1365-246X.2010.04884.x).

Schaeffer, A. J. and S. Lebedev. 2013. "Global Shear Speed Structure of the Upper Mantle and Transition Zone." *Geophysical Journal International* 194(1):417–49. Retrieved May 18, 2016 (http://gji.oxfordjournals.org/).

Stadler, Georg et al. 2010. "The Dynamics of Plate Tectonics and Mantle Flow: From Local to Global Scales." *Science* 329(5995).